# Measuring multi-spatiotemporal scale tourist destination popularity based on text granular computing

Chi Yunxian[1,2], Li Renjie [1,2]*, Zhao Shuliang[3], Guo Fenghua[4]

**1** College of Resources and Environment Science, Hebei Normal University, Shijiazhuang, Hebei, China, **2** Environmental Evolution and Ecological Construction Laboratory in Hebei Province, Shijiazhuang, Hebei, China, **3** College of Computer and Cyber Security, Hebei Normal University, Shijiazhuang, Hebei, China, **4** Institute of Geographical Sciences, Shijiazhuang, Hebei, China

* lrjgis@hebtu.edu.cn

**Data Availability Statement:** Data were purchased from Beijing Weimengkechuang network technology co. LTD (北京微梦科▪网▪技▪有限公司), which owns the commercial Sina microblog. The authors confirm that interested

## Abstract

User-generated content (UGC) is an important data source for tourism GIScience research. However, no effective approach exists for identifying hidden spatiotemporal patterns within multi-scale unstructured UGC. Therefore, we developed an algorithm to measure the tourist destination popularity (TDP) based on a multi-spatiotemporal text granular computing model, called TDPMTGC. To accurately granulate the spatial and temporal information of tourism text, tourism text data granules are used to represent landscape objects. These granules are unified objects that possess multiple attributes, such as spatial and temporal dimensions. The multi-spatiotemporal scales are characterized by the multi-hierarchical structure of granular computing, and transformations of granular layers and data granule size are achieved by scale selection in the spatial and temporal dimensions. Therefore, all scales between the spatial and temporal dimension are related, which allows for the comparability of the data granules of all spatial-spatial, temporal-temporal and spatial-temporal layers. This approach achieves a quantitative description and comparison of the popularity value of granules between adjacent scales and cross-scales. Therefore, the TDP with multi-spatiotemporal scales can be deduced and calculated in a systematic framework. We first introduce the conceptual framework of TDPMTGC to construct a quantitative measurement model of TDP at multi-spatiotemporal scales. Then, we present a dataset construction approach to support multi-spatiotemporal scale granular reorganization. Finally, TDPMTGC is derived to describe both the TDP at a single spatial or temporal scale and the patterns and processes of the TDP at multi-spatiotemporal scales. A case study from *Jiuzhaigou* shows that the TDP derived using TDPMTGC is consistent with the conclusions of existing studies. More importantly, TDPMTGC provides additional detailed characteristics, such as the contributions of different scenic spots in a tourist route or scenic area, the monthly anomalies and daily contributions of TDP in a specific year, the distinct weakening of tourist route scale in tourist cognition, and the daily variations of TDP during in-season and off-season times. This is the first time that a granular computing model has been introduced to tourism GIScience that provides a feasible scheme for reorganizing large-scale unstructured text

researchers can replicate their study findings in their entirety by directly obtaining the data from the third-party and following the protocol in our Methods section. Other researchers would be able to access the data set in the same manner as the authors, and the authors did not have any special access privileges that others would not have. The authors provide the following information about Sina microblog: located at Sina headquarters building, building 8, west district, no.10 Xibeiwang East Road, Haidian district, Beijing (北京市海淀区西北旺‧路10号院西区8号楼新浪‧部大厦); URL: https://open.weibo.com/wiki/C/2/place/nearby_timeline/biz.

**Funding:** Funded by 1. LRJ, grant number 41471127, Li Renjie, the National Natural Science Foundation of China, 2. LRJ, grant number D2015205208, Li Renjie, the Hebei Outstanding Youth Science Fund Cultivation Project, 3. CYX, grant number CXZZBS2018108, Chi Yunxian, the 2018 Hebei Province Doctoral Postgraduate Innovation Funding Project.

**Competing interests:** The authors have declared that no competing interests exist.

and constructing public spatiotemporal UGC tourism datasets. TDPMTGC constitutes a new approach for exploring tourist behaviors and the driving mechanisms of tourism patterns and processes.

# 1 Introduction

As an important part of geographical information science (GIScience [1]), tourism GIScience mainly studies a series of basic problems involved in processing, storing, extraction, management and analysis of tourism geographic information with computer technology. Tourist destination popularity (TDP), which refers to the tourists' attention to tourist destinations, is a popular issue in tourism GIScience research that can be expressed through the number of visitors [2–4], an index related to online searches and evaluations, and the user-generated content (UGC) published by tourists [5–7]. TDP is closely related to tourists' perceptions, preferences, and behaviors, which are critical to local tourism development because they provide important insights beyond the physical attributes of the landscape in a tourist destination and reflect its social significance [8]. By exploring the spatiotemporal characteristics [9–11] and evolutionary patterns of tourism destinations [12–13], researchers can analyze the influences of tourist perception [14], tourist satisfaction [15] and tourist spatiotemporal behaviors.

The focus of geographic spatiotemporal data mining is to study effective technical methods of exploring spatiotemporal data to support mining interesting patterns, anomalies and relationships within the data in the temporal and spatial dimensions [16]. While questionnaires are the traditional data acquisition approach for TDP [17–19], this approach has some deficiencies, such as small sample sizes, and it is difficult to guarantee the quality of survey results, which causes deviations in data analysis results [20]. Tourism big data provides a new solution to the above problems [21]. TDP can be analyzed using logs containing navigation, check-ins, and mobile positioning information. However, the semantic connotations of the TDP cannot be extracted due to the lack of content description. In the past decade, the quantity of information and the number of users on the Internet have increased tremendously; consequently, UGC is increasingly spreading through social networks. The multi-scale unstructured text-UGC usually contains spatiotemporal semantics; therefore, tourists can access this valuable information to choose tourist destinations or make travel plans. However, due to the explosive growth in the scale of such data, users must spend considerable time evaluating and extracting the collected information [22–23]. Therefore, mining knowledge from multi-scale unstructured UGC has become a popular research topic in various fields [24–27]. Scale is an important concept in geography [28–29]. Although the previous popularity analysis methods of tourist destination made multi-scale divisions on the temporal scale, they regarded a tourist destination as an integral unit on the spatial scale and often ignored its internal spatial characteristics, which affected the precision of the method. In-depth analysis of the spatiotemporal characteristics between scales helps improve model precision. However, establishing an accurate relationship between text and spatial units of different scales and integrating multi-spatial and multi-temporal scales into a systematic model are still obstacles in the study of tourism GIScience.

The granular computing (GrC) model can address the concept of scale well [28,30]. The GrC model divides the research object into several layers with different granularities, and each layer is interrelated to form one unified unit. Fine-grained information can support fine-scale

descriptions of scenic spots' popularity, location and spatiotemporal patterns, and the geographical laws of the larger scales can be analyzed by enlarging the granularity. We propose a model named tourist destination popularity measurement based on text granular computing (TDPMTGC). A GrC model is used to reconstruct the UGC data, granulate the tourism text [31], quantitatively describe the TDP, analyze the coupling relationship between different spatial and temporal scales [20], solve large-scale text data processing and complex problems [32–33], and describe the multi-scale geographic patterns and processes [34–35]. Determining the spatiotemporal behavior rules of tourist groups and analyzing the spatial patterns and driving mechanisms of TDP are both topics of considerable interest. TDPMTGC is extensible and can be applied to existing approaches or models to improve their detail. To accurately granulate the spatial and temporal information of tourism text, a tourism text data granule is used to represent a landscape object, which is a unified whole that possesses multiple attributes, such as spatial and temporal dimensions. The multi-spatiotemporal scales are characterized by the multi-hierarchical structure of GrC, and the transformations of granular layers and data granule size are realized by the scale selection in spatial and temporal dimensions. Therefore, all scales between the spatial and temporal dimension are related, which allows for the comparability of the data granules of all spatial-spatial, temporal-temporal and spatial-temporal layers. This approach achieves a quantitative description and comparison of the popularity value of granules between adjacent scales and cross-scales. Therefore, the TDP with multi-spatiotemporal scales can be calculated in a systematic framework. Thus, we can gain unique information by applying TDPMTGC to texts that cannot be obtained via other, possibly simpler, approaches (e.g., simply counting the number of visitors or the number of social media posts).

The main contributions of this paper to tourism GIScience are as follows. (1) We introduce the GrC model into tourism geography through the TDPMTGC algorithm, which constructs a quantitative model of TDP at multi-spatiotemporal scales based on GrC using the inclusion degree. The proposed TDPMTGC can describe the TDP at a single spatial or temporal scale as well as the patterns and processes of TDP at multi-spatiotemporal scales. (2) A dataset construction approach for the text GrC model is proposed to provide a feasible scheme for reorganizing large-scale unstructured text and constructing public spatiotemporal UGC tourism datasets. (3) The TDPMTGC model was successfully applied in the *Jiuzhaigou* area, resulting in some new insightful conclusions regarding TDP in this area. TDPMTGC provides a new data mining approach for exploring tourist behaviors and analyzing the driving mechanisms of tourism patterns and processes both spatially and temporally.

The remainder of this paper is organized as follows. The theory and approach of TDPMTGC are described in Section 2. In Section 3, we present the TDP computing method based on GrC. In Section 4, we report on a case study from *Jiuzhaigou* that demonstrates the feasibility of TDPMTGC. Finally, a discussion and conclusions are presented in Section 5.

## 2 Literature review

### 2.1 Semantic knowledge discovery in GIScience

The introduction of the concept of "Geographic Data Mining and Knowledge Discovery" [36] revealed that an important way to discover geographic laws in the big data era is to mine geographic data. Among this massive amount of data, more than 80% are composed of text, natural language, social media, etc. Many of these datasets exist in semi-structured or unstructured file formats, use different schema and lack geo-references or semantically meaningful links and descriptions of the corresponding geo-entities [37]. Therefore, how to mine the semantic features of spatiotemporal datasets has become an important topic in the field of GIScience. Among the possible uses are the application of social media data in tourism, geography and

other fields in humanities and social sciences. Mining the semantic information related to tourism destinations, cities, energy and so on is the main focus of current research which includes studies of TDP [5,38], tourist emotion [39], intention perception [40], areas of interest [41], travel trajectory [42], energy development [43] and so on. Moreover, due to the multi-scale characteristics of geographic datasets in both the spatial and temporal dimensions, it is necessary to mine and analyze the semantic knowledge at multi-spatiotemporal scales [44]. Building a mathematical model is an effective method for discovering semantic knowledge in GIScience. The application of ontologies, Bayesian networks and other models has laid the foundation for multi-spatiotemporal semantic mining [45–46].

The TDPMTGC method proposed in this paper adopts social media data, which can reflect users' real intentions, to construct a text GrC model that can quantitatively calculate TDP at multi-spatiotemporal scales allowing it to mine semantic knowledge concerning TDP from UGC text.

## 2.2 Spatiotemporal data mining

Geographic spatiotemporal big data include both earth observation and human behavior data [47] whose value is reflected in hidden rules and knowledge [48–50]. Among these data types, the objects of earth observations are the earth's surface elements, and the data that can be mined include satellite remote sensing data, monitoring station data, UAV images, and so on [51–52]. With location at their core, these types of data can be structured easily and represented by positional space [47]. In contrast, the main body of human behavior big data concerns human beings, and the data that can be mined include social media data [5,38], mobile phone signals [42], taxi route [53], and so on. The structural types in such data are complex and diverse and can be represented by flow spaces (including people flow, information flow, relationship flow, etc.) [16,47,54].

The *Sina* microblog adopted as the data source in this paper belongs to the human behavior big data type. Mining TDP in both the spatial and temporal dimensions can reveal tourists' attention to tourist destinations from multi-spatiotemporal scales.

## 2.3 Popularity analysis of tourist destination

In the big data era, the volume of social media data that reflects real user preferences and the wisdom of crowds has undergone explosive growth. Integrating such big data with rich spatiotemporal information and semantics is an effective way to find popular routes, scenic spots, etc. [38]. Such knowledge can provide references that tourism managers can use to plan tourism resources and that tourists can use to plan reasonable itineraries and improve their tourism experiences. Analyzing tourists' online comments and microblog posts can reveal their perceptions and preferences regarding a tourism destination which can then be used to reflect the TDP [22,55]. In addition, tourism destination recommender systems can help users cope with information overload, provide personalized recommendations and services, and help find popular tourist destinations [23,41,56].

The above methods take tourism destinations as a complete spatial unit when analyzing their popularity at multi-temporal scales. The method proposed in this paper divides space and time into multiple scales and then integrates them into a systematic framework to achieve a fine-grained multi-scale popularity analysis.

## 2.4 Granular computing model

The concept of "scale" in the field of GIScience can be modeled using the hierarchical structure of granular computing. Data granules are formal entities that facilitate a way of

organizing knowledge about data and relationships. GrC is concerned with the development and processing of data granules [57]. Representing geographic objects as data granules can effectively identify spatiotemporal scales and periodic patterns and improve the logicality, systematicity and efficiency of decision-making [58–59]. GrC makes it possible to flexibly adjust levels and make deductions between levels [57]. By establishing a systematic hierarchical framework [60], the evaluation results from a previous granular level can be employed as criteria in at a subsequent granular level [61], allowing locally constructed models to be used to deduce the global model [62].

To our knowledge, this paper is the first to introduce a GrC model to tourism GIScience and expand its application in GIScience. We use tourism text data granules to represent the landscape objects in tourism GIScience and depict the multi-spatiotemporal scales in tourism GIScience through the multi-hierarchical structure of GrC. Then, the multi-spatiotemporal scales TDP can be deduced within a systematic framework.

## 3 Theory and method

Mathematical modeling is used to build theoretical models that reflect real problems; then, solving the model can yield results that are also the solution to the real problem. In the big data era, scholars in the field of GIScience use rich social media resources [63] and mathematical methods to model complex GIScience problems [64–65] and explore the patterns and motivations of human activities at multi-spatiotemporal scales [66–68].

The existing approaches calculate the TDP through two methods. The first method uses mobile tourist locations to determine popularity. This approach has high spatial accuracy but lacks the semantic content of multi-spatial scales. The second method determines popularity through keywords in tourist UGC. This approach has clear semantic content but insufficient spatial accuracy. Given the problems with above methods, we introduce the TDPMTGC algorithm, which uses the inclusion degree based on conditional probability for GrC. Text data granules are applied to calculate the TDP to achieve quantitative descriptions and in-depth mining of the multi-spatiotemporal TDP. The design idea of this method is as follows. We first introduce the concept of 'information granule' into tourism GIScience to design the text granulation method and the data granular structure of tourism UGC with a spatiotemporal scale. Then, we introduce the concept of inclusion degree based on conditional probability. Finally, we present the TDPMTGC conceptual framework using inclusion degree. TDPMTGC is used to describe both TDP at a single spatial or temporal scale and the patterns and processes of TDP at multi-spatiotemporal scales.

### 3.1 Definitions of related concepts

GrC originated from the idea of 'information granulation', which was first introduced by Zadeh [30] in 1979. Information from different perspectives and levels is defined as different granules that can be used for massive data processing and complex problem solving [32]. GrC provides an effective theory and method for solving the thematic organization problem of big data [33]. Introducing GrC into tourism GIScience also generates the related concepts of GrC in tourism texts.

**Definition 1. Tourism Text Data Granule**. A tourism text block defined based on tourism text elements characterized by time, space, similarity, adjacency, uncertainty, or function [69–70], denoted as *Gr*.

**Definition 2. Tourism Text Data Granulation**. The granulation process that divides large-scale and complex tourism text into small and semantically clear tourism text data granules

based on a set of criteria associated with spatial and temporal scale features or other geographical thematic semantics.

**Definition 3. Tourism Text Data Granular Layer**. A layer composed of a set of tourism text data granules based on certain granulation criteria, denoted as *L*.

**Definition 4. Multi-scale Tourism Text Data Granular Structure**. The geographical relational structure formed by the connections between multiple tourism text data granules corresponding to different granulation criteria, denoted as *GrS*.

**Definition 5. Tourism Text Data Granular Computing**. Computing processes that use tourism text data granules to describe, analyze, and solve tourism text mining problems from different scales and perspectives [32].

## 3.2 Granulation criteria of tourism text

### 3.2.1 Representation of tourism text data granules.

Geographical objects can be described via four dimensions: longitude, latitude, altitude and time. The first three dimensions constitute the space (longitude, latitude, height). Both time and space have scalar properties [28] and are related.

1. **Granule**: Tourism text data granules are represented as $Gr_j^{S_r \& T_r}$, where $S_r$ and $T_r$ represent the spatial and temporal scales of the data, respectively. $S_r = \{1, \cdots, S_{\max}\}$ and $T_r = \{1, \cdots, T_{\max}\}$, where $S_{\max}$ and $T_{\max}$ are the total number of spatial and temporal granular layers, respectively, and *j* is an index reflecting the order of granules. Tourism text data granules are the basic elements of tourism text data GrC models.

2. **Spatial and temporal granular layers**: A tourism text data granular layer is an abstract representation of tourism GIScience problems. Different spatial and temporal scales can be described as different spatial and temporal granular layers.
   As shown in Fig 1, *SRuleSet* is a set of spatial granulation criteria. The criteria for each layer are $SRule^{s \& T_r} \subseteq SRuleSet(s = 1, \cdots, S_{\max} - 1)$. Each granule $Gr_j^{s \& T_r}$ in layer *s* is granulated into $N^{(s+1)\&T_r}$ granules $Gr_j^{(s+1)\&T_r}(s = 1, \cdots, S_{\max} - 1; j = 1, 2, \cdots, N^{(s+1)\&T_r})$ in layer $s + 1$ with $SRule^{s \& T_r}$. All the granules located in layer $s + 1$ constitute the tourism text data granular layer. The granules that satisfy the conditions of $Gr_j^{s \& T_r} \in L^{s \& T_r}(s = 1, \cdots, S_{\max}; j = 1, 2, \cdots, N^{s \& T_r})$ are in the same layer and usually have geospatial semantic properties that conform to the same theme or objective criteria. Similarly, granulation criteria rules $TRule^{S_r \& t} \subseteq TRuleSet(t = 1, \cdots, T_{\max} - 1)$ apply to temporal scales.

3. **Granular spatial and temporal structures**: Each spatial scale corresponds to a spatial granular layer $L^{s \& T_r}(s = 1, \cdots, S_{\max})$, and the multi-spatial granular structure is composed of spatial granular layers using a spatial granulation criteria, $SRule^{s \& T_r}$. Similarly, a multi-temporal granular structure is composed of spatial granular layers $L^{S_r \& t}(t = 1, \cdots, T_{\max})$. Taking *Jiuzhaigou* for example, for the 4-layer spatial granular structure of 'tourism destinations→scenic areas→tourist routes→scenic spots', the granule that represents *Rizegou* in the 3rd layer can be granulated into many scenic spot granules (belonging to *Rizegou*) in the 4th layer using the granulation criteria 'tourist routes→scenic spots'.

### 3.2.2 Granular structure of tourism text data.

1. **The structure of a tourism text data granule**. A tourism text data granule is a complete entity with multiple attributes, such as space and time, which must be described from spatial, temporal and other dimensions. Among them, both the spatial and temporal

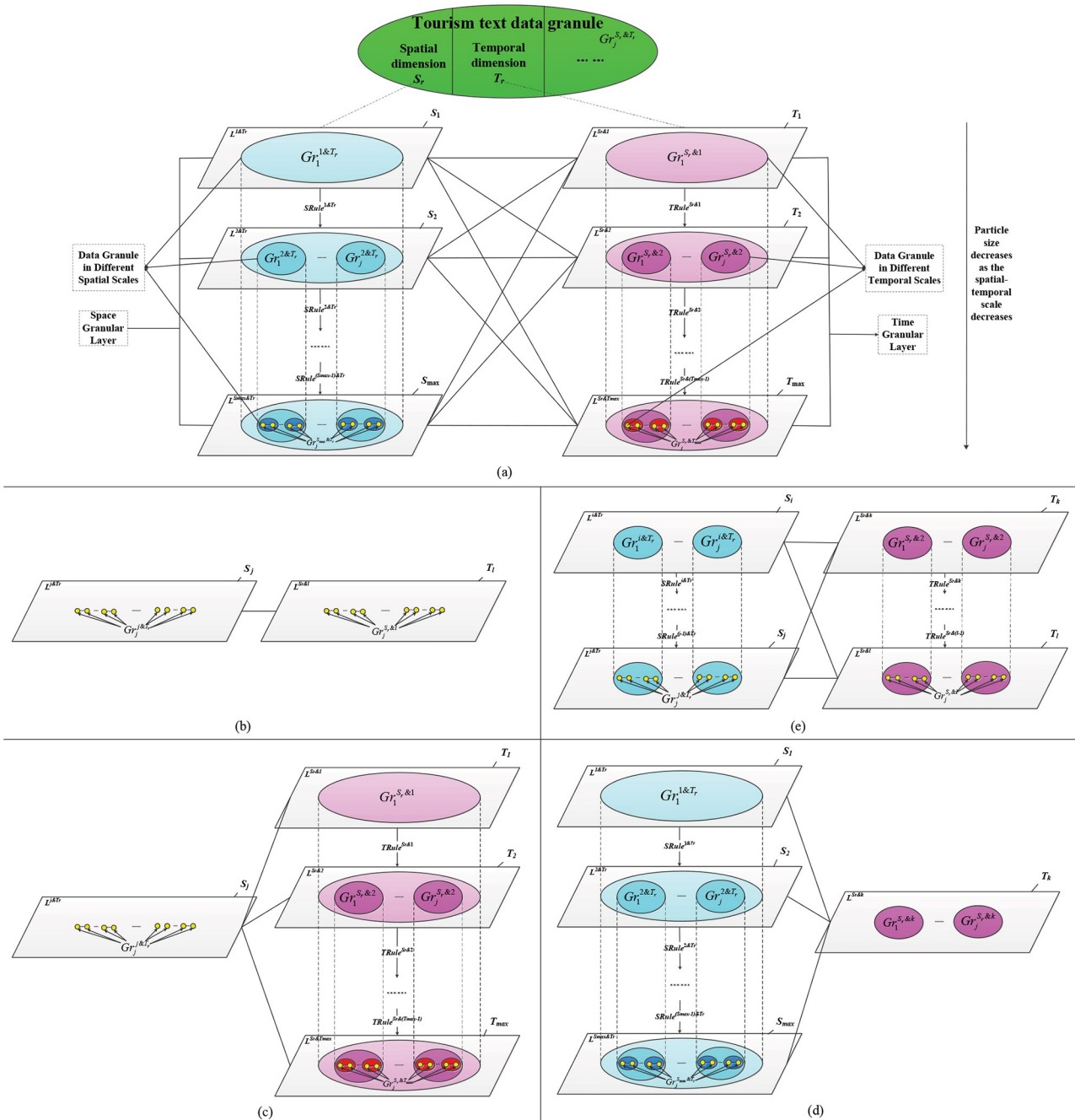

**Fig 1. Schematic diagram of the tourism text data granular structure at multi-spatiotemporal scales.** (a) granular structure of tourism text data; (b) popularity mining of single spatial scale data granules at a single temporal scale; (c) popularity mining of single spatial scale data granules at multi-temporal scales; (d) popularity mining of multi-spatial scale data granules at a single temporal scale; (e) popularity mining of multi-spatial scale data granules at a multi-temporal scale.

dimensions contain multiple scales; thus, the multi-scale structure of granules corresponds to these multi-spatiotemporal scales (see Fig 1(a)). Using this approach, the time and space dimensions are integrated into a single systematic model reflected as attributes of data granules. To describe the spatiotemporal characteristics of the data granules, it is

necessary to clearly indicate their spatiotemporal scale, which can be divided into the following situations: ① To describe the characteristics of data granules at a particular spatiotemporal scale, it is necessary to fix the spatial and temporal scales of the granules (see Fig 1(b)); ② To describe the characteristics of data granules at a specific spatial (or temporal) scale, it is necessary to fix the spatial (or temporal) scale of the granules and mine the evolution rules of granules at that multi-temporal (or multi-spatial) scale (see Fig 1(c) and 1(d)); and ③ To describe the characteristics of data granules at multi-spatiotemporal scales, multiple scales of the spatial and temporal dimensions of the granules should be selected to perform comprehensive mining (see Fig 1(e)).

2. **The implementation method of multi-spatiotemporal scale granular structure**. The multi-spatiotemporal scale granular structure of tourism text data is represented by the complete graph shown in Fig 1(a), in which layers $1 \sim S_{\max}$ of the multi-spatial granular structure correspond to the $S_{\max}$ scales. The data granules in the upper scale are transformed into those in the lower scale using the granulation criteria $SRule^{s \& T_r}(s = 1, \cdots, S_{\max}-1)$. The data granules decrease as the scale decreases. Similarly, layers $1 \sim T_{\max}$ of the multi-temporal granular structure correspond to the $T_{\max}$ scales, and granules in the upper scale are transformed into those in the lower scale using the granulation criteria $TRule^{S_r \& t}(t = 1, \cdots, T_{\max}-1)$. A complete graph represents the existence of an edge (i.e., a correlation) between any spatial-spatial, temporal-temporal, or spatial-temporal scales. There are $S_{\max}(S_{\max}-1)/2$ edges among the spatial-spatial scales, $T_{\max}(T_{\max}-1)/2$ edges among the temporal-temporal scales, and $S_{\max} \cdot T_{\max}$ edges among the spatial-temporal scales; thus, the total number of edges is $(S_{\max}+T_{\max})(S_{\max}+T_{\max}-1)/2$. The correlation between temporal scales is presupposed by the "spatial-temporal" correlation (i.e., the correlation between two temporal scales '$T_k$—$T_l$' for a spatial scale $S_i$ is obtained by granulating $S_i$ in layers $T_k$ and $T_l$, which yields the correlations '$S_i$—$T_k$' and '$S_i$—$T_l$'). The granular structure of tourism text data can be used not only to mine features of small-scale landscapes (where $S_1$ represents a tourist destination) over a short period (such as when $T_1$ represents an annual scale) but also to mine the life cycle evolutionary laws at large scales (where $S_1$ represents a national or even a global scale) over long periods (such as when $T_1$ represents several centuries (if the data are available)). According to the actual needs, subgraphs can be extracted from Fig 1(a) to achieve landscape law mining at a single-space/single-time scale (see Fig 1(b)), single-space/multiple-time scales (see Fig 1(c)), multiple-space/single-time scales (see Fig 1(d)), and multiple-space/multiple-time scales (see Fig 1(e)).

In conclusion, a tourism text data granule is a unified whole possessing multiple attributes, such as a spatial and a temporal dimension. The transformations of granular layers and data granule size are achieved by scale selection in both the spatial and temporal dimensions. Therefore, all the scales between spatial and temporal dimension are related, which allows for the comparability of the data granules of all spatial-spatial, temporal-temporal and spatial-temporal layers. This approach allows for comparisons of the popularity value of data granules both among adjacent scales and across scales, forming unique information that we can gain by applying TDPMTGC to texts that cannot be obtained via other, possibly simpler, approaches (e.g., simply counting the number of visitors or the number of social media posts). We can analyze the geographic spatiotemporal relations among the multiple granular layers using the granular structure $GrS$. Thus, $GrS$ is a useful tool for finely describing the multi-spatiotemporal patterns of TDP.

### 3.3 Multi-spatiotemporal tourism text granular computing model based on inclusion degree

The approach related to GrC include fuzzy sets [71], neighborhood topology [72], inclusion degree [73], formal concept analysis [74], algebraic lattices [75], calculus [76], and logical views [77]. Inclusion degree theory is one of the classic methods for implementing GrC [31–32,78]. Inclusion degree theory contains all the results of uncertain reasoning [32]. Conditional probability is a form of uncertain reasoning, representing one kind of inclusion degree. Uncertainty exists in the multi-spatiotemporal features of tourism text data. For example, the text 'Wucaichi' is unclear because it does not specify the scenic area or tourist route to which the scenic spot belongs. The inclusion degree is generated based on conditional probability to quantitatively infer TDP at multi-spatiotemporal scales with different granularities, analyze the couplings of TDP between different spatial and temporal scales, and mine the evolutionary patterns of TDP.

**3.3.1 Definition of inclusion degree.** **Definition 6. Inclusion Degree**. Let $DSet$ be a tourism text dataset and assume that $DSet$ has three subsets $D_A, D_B, D_C \subseteq DSet$. If $ID(D_B / D_A)$ exists and satisfies the following three properties

1. Nonnegative: $0 \leq ID(D_B/D_A) \leq 1$;

2. Normative: $ID(D_B/D_A) = 1$, when $D_A \subseteq D_B$;

3. Transitive: $ID(D_A/D_C) \leq ID(D_A/D_B)$, when $D_A \subseteq D_B \subseteq D_C$,

then $ID(D_B/D_A)$ is the inclusion degree to which $D_B$ contains $D_A$ (or $D_A$ is contained in $D_B$).

**3.3.2 Generating the inclusion degree using conditional probability.** The following formulas are appropriate for both spatial and temporal dimensions.

**Theorem 1**. Let $X$ be a finite set for which $Gr_A$ and $Gr_B$ meet the condition that $Gr_A, Gr_B \subseteq X$, and $N(Gr_A)$ is the number of elements in $Gr_A$. Then $N(Gr_A)/n$ is the corresponding probability measure. If $P$ is the probability distribution of $X$, then

$$ID(Gr_B/Gr_A) = \frac{N(Gr_A \text{ I } Gr_B)}{N(Gr_A)} = \frac{P(Gr_A \text{ I } Gr_B)}{P(Gr_A)} = P(Gr_B|Gr_A) \tag{1}$$

is the inclusion degree of $X$, when $P(Gr_A) > 0$.

**3.3.3 The inclusion degree of multi-spatiotemporal tourism text data granularity.**
**Property 1**. If the granules of each spatial granular layer $Gr_j^{s\&T_r} \in L^{s\&T_r}(s = 1, \cdots, S_{\max}; j = 1, 2, \cdots, N^{s\&T_r})$ meet any of the following conditions

1. A data granule in one spatial granular layer is a subset of that in the upper granular layer, i.e., $Gr_j^{s\&T_r} \subseteq Gr_j^{(s-1)\&T_r}$.

2. The collection of several data granules in one spatial granule is a subset of that in the upper granule, i.e., $\sum Gr_j^{s\&T_r} \subseteq Gr_j^{(s-1)\&T_r}$.

then the inclusion degree of the tourism text data granules of two adjacent spatial scales can be defined as $ID\left(Gr_j^{(s-1)\&T_r}/Gr_j^{s\&T_r}\right)$ (or $ID\left(\left(\sum Gr_j^{s\&T_r}\right)/Gr_j^{s\&T_r}\right)$), which is called the inclusion degree to which $Gr_j^{(s-1)\&T_r}$ contains $Gr_j^{s\&T_r}$ (or $\sum Gr_j^{s\&T_r}$ contains $Gr_j^{s\&T_r}$).

**Property 2**. If the granules of each temporal granule layer $Gr_j^{S_r\&t} \in L^{S_r\&t}(t = 1, \cdots, T_{\max}; j = 1, 2, \cdots, N^{S_r\&t})$ meet any of the following conditions

1. A data granule in one temporal granular layer is a subset of that in a higher layer, i.e., $Gr_j^{S_r \& t} \subseteq Gr^{S_r \& (t-1)}$.

2. A collection of several data granules in one temporal granular layer is a subset of that in a higher layer, i.e., $Gr_j^{S_r \& t} \subseteq \sum Gr_j^{S_r \& t} \subseteq Gr_j^{S_r \& (t-1)}$.

then the inclusion degree of a tourism text data granules at two adjacent temporal scales can be defined as $ID(Gr^{S_r \& (t-1)}/Gr^{S_r \& t})$ (or $ID\left(\left(\sum Gr_j^{S_r \& t}\right)/Gr_j^{S_r \& t}\right)$). It is called the inclusion degree in which $Gr^{S_r \& (t-1)}$ contains $Gr_j^{S_r \& t}$, (or $\sum Gr_j^{S_r \& t}$ contains $Gr_j^{S_r \& t}$).

**3.3.4 Multi-spatiotemporal tourism text granular computing model based on inclusion degree.** Combining Sect. 3.3.2 and Sect. 3.3.3, the multi-spatial tourism text GrC model based on inclusion degree can be written as

$$ID\left(Gr_j^{s \& T_r}/Gr_j^{(s-1)\& T_r}\right) = P\left(Gr_j^{s \& T_r}|Gr_j^{(s-1)\& T_r}\right) = P\left(Gr_j^{s \& T_r} \mathrm{I} \, Gr_j^{(s-1)\& T_r}\right)/P\left(Gr_j^{(s-1)\& T_r}\right), \quad (2)$$

or

$$ID\left(Gr_j^{s \& T_r}/\left(\sum Gr_j^{s \& T_r}\right)\right) = P\left(Gr_j^{s \& T_r}|\left(\sum Gr_j^{s \& T_r}\right)\right) = P\left(Gr_j^{s \& T_r} \mathrm{I} \left(\sum Gr_j^{s \& T_r}\right)\right)/P\left(\sum Gr_j^{s \& T_r}\right). \quad (3)$$

Similarly, the multi-temporal tourism text GrC model based on inclusion degree is as follows

$$ID\left(Gr^{S_r \& t}/Gr^{S_r \& (t-1)}\right) = P\left(Gr^{S_r \& t}|Gr^{S_r \& (t-1)}\right) = P\left(Gr^{S_r \& t} \mathrm{I} \, Gr^{S_r \& (t-1)}\right)/P\left(Gr^{S_r \& (t-1)}\right), \quad (4)$$

or

$$ID\left(Gr_j^{S_r \& t}/\left(\sum Gr_j^{S_r \& t}\right)\right) = P\left(Gr_j^{S_r \& t}|\left(\sum Gr_j^{S_r \& t}\right)\right) = P\left(Gr_j^{S_r \& t} \mathrm{I} \left(\sum Gr_j^{S_r \& t}\right)\right)/P\left(\sum Gr_j^{S_r \& t}\right). \quad (5)$$

# 4 The tourist destination popularity computing approach based on granular computing model

To implement TDPMTGC to support the quantitative calculation of TDP at multi-spatiotemporal scales, we first need to organize and construct a standard dataset that meets the requirements of the granular computing model. Then, we design the TDP computing approach based on the GrC model.

## 4.1 The granular computing model dataset construction approach

The spatial information (such as toponymy) in multi-scale unstructured UGC data is implicit in the text and needs to be identified layer by layer. Moreover, the data granules in the lower layer are subsets of those in the next highest layer and a number of cross-scale layers (i.e., tourist route granules at a tourist route scale not only include single spot granules but also single route with multiple spots granules and multiple routes with multiple spots granules at the scenic spot scale. Similarly, they include single-route and multiple-route granules at a tourist route scale). After completing the construction of granules in the lower layer, they can be directly integrated into the granules in the upper layer, thus expanding to larger granules layer by layer. Because of this inclusion relationship between scales in the spatial dimension, the dataset is constructed from bottom to top using a scale from small to large and a granular scale that moves from fine to coarse. The temporal information in UGC data is explicit in each text; thus, data granules in lower layers inherit the labels of those in the upper layers (for example, a

granule at a monthly scale must belong to a certain granule at a yearly scale). We adopt a tree structure to complete the construction of the data granules in the upper layer and then decompose them downward layer by layer. This approach clearly indicates the inheritance relationship among the data granules of each layer. Hence, in the temporal dimension, based on the spatial dataset, the dataset is constructed from top to bottom using a scale from large to small and a granular scale that moves from coarse to fine.

Consequently, we use the granular structure of tourism text data described in section 3.2.2 to construct datasets that reflect the spatial and temporal dimensions, respectively. Common spatial scales are implemented in tourist GIScience, such as scenic spots, tourist routes, scenic areas, tourist destinations, provinces, nations, etc. Similarly, common temporal scales are implemented, such as year, month, week, day, hour, minute, and second. The number of spatial and temporal scales should be selected according to the size of the tourist destination (i.e., smaller scenic areas can skip the tourist route scale). In this paper, we use four scales in the spatial dimension, namely, "scenic spot—tourist route—scenic area—tourist destination", and four scales in the temporal dimension, namely, "year—month—day—time", as examples to introduce the dataset construction method of spatial and temporal dimension.

**4.1.1 Dataset construction in the spatial dimension.** The UGC texts related to tourist destinations are composed of two parts: text that mentions toponym features of a scenic area at different spatial scales (i.e., toponym text) and texts that do not contain any scale-related toponym features (i.e., nontoponym text). Text selection is conduct to discover toponym features at different spatial scales, such as scenic spots, tourist routes, scenic areas, and tourist destinations. Starting from the smallest granule, namely, a scenic spot, text datasets of scenic spots, tourist routes, scenic areas, and tourist destinations are successively established (see Fig 2).

1. Scenic spot scale $L^{4\&T_r}$: the text collection of each scenic spot, namely, a scenic spot granule $Gr^{4\&T_r}_{line\_j}$ ($line \in \{A,B,\cdots,X\}$ represents the name of a tourist route to which scenic spot $j$ belongs), is filtered by the name of the scenic spot. The filtering results are divided into three categories: 1) a single spot, expressed as $1 - S^{4\&T_r}$ (for simplicity, $A^4$ is used instead of $1 - S^{4\&T_r}$ in the following passage), each of which describes a single scenic spot; 2) a single route with multiple spots, expressed as $1 - R\_m - S^{4\&T_r}$ ($B^4$ for short), each of which describes several scenic spots along a single tourist route; 3) multiple routes with multiple spots, expressed as $m - R\_m - S^{4\&T_r}$ ($C^4$ for short), which describes multiple scenic spots along multiple routes simultaneously. Among these, the $A^4$ granules of a tourist route can be summed to describe a complete set of individual scenic spot texts for a tourist route, satisfying $Gr^{3\&T_r}_{line\odot} = \sum_{j \in line} Gr^{4\&T_r}_{line\odot\_j}$. $B^4$ granules do not have additivity (because a text that contains multiple scenic spots is counted multiple times). $Gr^{4\&T_r}_{line*\_j}$ refers to the number of times that scenic spot $j$ appears in $Gr^{4\&T_r}_{line*}$, satisfying $Gr^{4\&T_r}_{line*\_j} \subseteq Gr^{3\&T_r}_{line*}$ ($j \in line$). $C^4$ granules are also not additive. $Gr^{4\&T_r}_{\&\_line\_j}$ refers to the number of times that scenic spot $j$ appears in $Gr^{3\&T_r}_{\&\_line}$, which satisfies $Gr^{4\&T_r}_{\&\_line\_j} \subseteq Gr^{3\&T_r}_{\&\_line}$ ($j \in line$). Each scenic spot meets the requirement $Gr^{4\&T_r}_{line\_j} = Gr^{4\&T_r}_{line\odot\_j} \bigcup Gr^{4\&T_r}_{line*\_j} \bigcup Gr^{4\&T_r}_{\&\_line\_j}$. For example, one text is included in three scenic spots that are attached to three routes, namely, *Xiniuhai* (belonging to *Shuzhenggou*), *Nuorilangpubu* (belonging to *Rizegou*), and *Wucaichi* (belonging to *Zechawagou*), which is a $C^4$ and is counted three times during the calculation of scenic spot popularity. Therefore, the sum of the scenic spots in $C^4$ is greater than the number of time they originally appear in the texts.

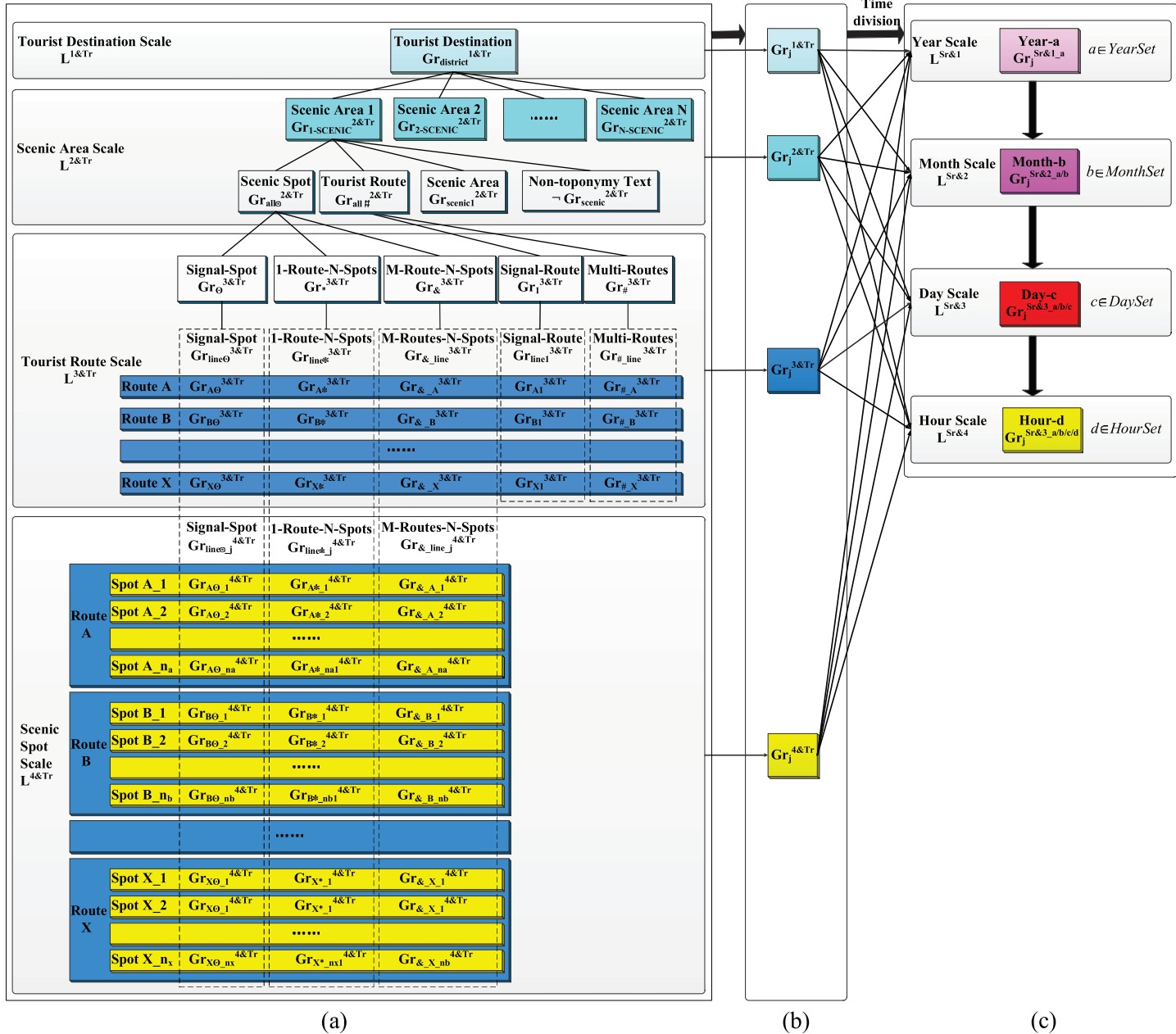

**Fig 2. Construction of multi-spatiotemporal scale datasets of tourism text based on granular computing.** (a) dataset construction in the spatial dimension; (b) data granules at different scales; (c) dataset construction in the temporal dimension. The dataset in Fig 2(a) is constructed from bottom to top in four scales, including scenic spot, tourist route, scenic area and tourist destination. Scenic spot granules at the scenic spot scale are colored yellow; and each scenic spot granule represents a scenic spot composed of three parts: a single spot, a route with multiple spots and multiple routes with multiple spots. Route granules at the tourist route scale are colored blue, and each route granule represents a route composed of five parts: single spot, one route with multiple spots, multiple routes with multiple spots, single route and multiple routes. The single spots are represented by the complete set of scenic spots in the route (i.e., union). Similarly, the same approach is used for representing one route with multiple spots and multiple routes with multiple spots. Scenic area granules at the scenic area scale are colored light blue, and each scenic area granule represents a scenic area composed of four parts: scenic spot, tourist route, scenic area and nontoponym text. Among these, scenic spots are represented by the union set of single spots, one route with multiple spots and multiple routes with multiple spots at the tourist route scale, and a tourist route is composed of the union set of single routes and multiple routes at the tourist route scale. The tourist destination granules within the scale of tourist destination are colored aqua, and each tourist destination granule represents a tourist destination, which is represented by the union of several scenic areas within the tourism destination. The dataset in Fig 2(c) is constructed using four scales—year, month, day and hour—in top-down order, and the corresponding colors are light pink, dark pink, red and yellow. The granules at each spatial scale in Fig 2(b) are divided into four temporal scales.

2. Tourist route scale $L^{3\&T_r}$: a tourist route granule $Gr_{line}^{3\&T_r}$ is made up of scenic spots $Gr_{all\odot}^{2\&T_r}$ and tourist routes $Gr_{all\#}^{2\&T_r}$. Among these, $Gr_{all\odot}^{2\&T_r}$ corresponds to $1 - S^{3\&T_r}$, $1 - R\_m - S^{3\&T_r}$, and $m - R\_m - S^{3\&T_r}$ ($A^3$, $B^3$ and $C^3$ for short) at the scenic spot scale, satisfying $Gr_{all\odot}^{2\&T_r} = Gr_{line\odot}^{3\&T_r} \bigcup Gr_{line*}^{3\&T_r} \bigcup Gr_{\&\_line}^{3\&T_r}$. The tourist routes $Gr_{all\#}^{2\&T_r}$ include both texts describing tourist routes separately (referred to as single route and expressed as $1 - R^{3\&T_r}$($D^3$ for short)), and those describing multiple routes at the same time (referred to as multiple routes and expressed as m $- R^{3\&T_r}$($E^3$ for short)), satisfying $Gr_{all\#}^{2\&T_r} = Gr_{line1}^{3\&T_r} \bigcup Gr_{line\#}^{3\&T_r}$. Each tourist route meets the requirements $Gr_{line}^{3\&T_r} = Gr_{line\odot}^{3\&T_r} \bigcup Gr_{line*}^{3\&T_r} \bigcup Gr_{\&\_line}^{3\&T_r} \bigcup Gr_{line1}^{3\&T_r} \bigcup Gr_{\#\_line}^{3\&T_r}$.

3. Scenic area scale $L^{2\&T_r}$: a toponym text granule $Gr_{scenic}^{2\&T_r}$ of a scenic area includes both scenic spots $Gr_{all\odot}^{2\&T_r}$ and tourist routes $Gr_{all\#}^{2\&T_r}$ at the tourist route scale, and descriptions of single scenic areas. $Gr_{scenic1}^{2\&T_r}$ (referred to as scenic area), satisfying $Gr_{scenic}^{2\&T_r} = Gr_{all\odot}^{2\&T_r} \bigcup Gr_{all\#}^{2\&T_r} \bigcup Gr_{scenic1}^{2\&T_r}$.

4. Tourist destination scale $L^{1\&T_r}$: a tourist destination granule $Gr_{district}^{1\&T_r}$ contains granules of each scenic area within the tourist destination, $Gr_{j-SCENIC}^{2\&T_r}$, satisfying $Gr_{district}^{1\&T_r} = \bigcup_{j=1}^{N} Gr_{j-SCENIC}^{2\&T_r}$.

   Each scenic area granule contains both toponym text granules $Gr_{scenic}^{2\&T_r}$ and nontoponym text granules $Gr_{scenic}^{2\&T_r}$, satisfying $Gr_{SCENIC}^{2\&T_r} = Gr_{scenic}^{2\&T_r} \bigcup Gr_{scenic}^{2\&T_r}$.

The schematic diagram in Fig 2 shows how the granules located in each spatial granular layer are consistent with the colors of the corresponding granules in Fig 1.

**4.1.2 Dataset construction in the temporal dimension.** The granules at each spatial scale are granulated from top to bottom according to the temporal scale, and four temporal scale granules (year, month, day and hour) are obtained:

$Gr_j^{S_r\&1-a} \rightarrow Gr_j^{S_r\&2-a/b} \rightarrow Gr_j^{S_r\&3-a/b/c} \rightarrow Gr_j^{S_r\&4-a/b/c/d}$, satisfying

$$Gr_j^{S_r\&1} = \sum_{a \in YearSet} Gr_j^{S_r\&1-a} = \sum_{a \in YearSet} \sum_{b \in MonthSet} Gr_j^{S_r\&1-a/b}$$

$$= \sum_{a \in YearSet} \sum_{b \in MonthSet} \sum_{c \in DaySet} Gr_j^{S_r\&1-a/b/c} = \sum_{a \in YearSet} \sum_{b \in MonthSet} \sum_{c \in DaySet} \sum_{d \in HourSet} Gr_j^{S_r\&1-a/b/c/d}, \quad (6)$$

where $S_r$ represents the scales of tourist destinations, scenic areas, tourist routes and scenic spots.

## 4.2 Quantifying tourist destination popularity based on granular computing

**4.2.1 Tourism destination popularity at different spatial scales.** A tourism text dataset organized based on the spatial dimension can support the calculation of TDP at the four spatial scales of scenic spots, tourist routes, scenic areas, and tourist destinations and can be used to explore the coupling relationships between tourist behavior and spatial semantics at different scales. The following is a detailed description of the approaches used to calculate TDP at different spatial scales.

**(I) Scenic spot scale**. The popularity of each scenic spot is calculated based on its tourist route, which is contributed to jointly by the popularity of $A^4$, $B^4$ and $C^4$. The total popularity

of scenic spots is

$$ID\left(Gr_{line\_j}^{4\&T_r}/Gr_{all-spot}^{4\&T_r}\right) = P\left(Gr_{line\_j}^{4\&T_r}|Gr_{all-spot}^{4\&T_r}\right) = P\left(Gr_{line\_j}^{4\&T_r}\right)/P\left(Gr_{all-spot}^{4\&T_r}\right). \tag{7}$$

Among these, $Gr_{line\_j}^{4\&T_r} = Gr_{line\odot\_j}^{4\&T_r} \bigcup Gr_{line*\_j}^{4\&T_r} \bigcup Gr_{\&\_line\_j}^{4\&T_r}$, and $Gr_{all-spot}^{4\&T_r} = \bigcup\limits_{line\in\{A,B,\cdots,X\}} Gr_{line\_j}^{4\&T_r}$

which includes

$$A^4 : ID\left(Gr_{linee\_j}^{4\&T_r}/Gr_{all-spot}^{4\&T_r}\right) = P\left(Gr_{linee\_j}^{4\&T_r}|Gr_{all-spot}^{4\&T_r}\right) = P\left(Gr_{linee\_j}^{4\&T_r}\right)/P\left(Gr_{all-spot}^{4\&T_r}\right),$$

$$B^4 : ID\left(Gr_{line*\_j}^{4\&T_r}/Gr_{all-spot}^{4\&T_r}\right) = P\left(Gr_{line*\_j}^{4\&T_r}|Gr_{all-spot}^{4\&T_r}\right) = P\left(Gr_{line*\_j}^{4\&T_r}\right)/P\left(Gr_{all-spot}^{4\&T_r}\right),$$

$$C^4 : ID\left(Gr_{\&\_line\_j}^{4\&T_r}/Gr_{all-spot}^{4\&T_r}\right) = P\left(Gr_{\&\_line\_j}^{4\&T_r}|Gr_{all-spot}^{4\&T_r}\right) = P\left(Gr_{\&\_line\_j}^{4\&T_r}\right)/P\left(Gr_{all-spot}^{4\&T_r}\right).$$

For example, there are 49 scenic spots, and the set of scenic spot granules is a subset of the "scenic spots' at the tourist route scale. Because the numbers of scenic spots in $B^4$ and $C^4$ are greater than 1, the same text is counted using the number of actual scenic spots in the text. Taking "*Zhenzhutanpubu*" as an example, the number of $A^4$ is 449, and the total number of texts describing scenic spots is 10,607 (that is, the sum of $A^4$, $B^4$ and $C^4$ for the 49 scenic spots). Hence, the popularity of $A^4$ of *Zhenzhutanpubu* is 4.23%.

Because each text describing $B^4$ and $C^4$ contains multiple scenic spots, the same text is counted repeatedly in multiple scenic spot granules, which leads to values larger than the actual occurrences of scenic spot texts that are contained in $Gr_{all-spot}^{4\&T_r}$. Therefore, the distribution characteristics of different parts of scenic spot granules within the scope of the scenic area are considered, and the comprehensive popularity of scenic spots within the scope of a scenic area can be obtained:

$$\begin{aligned}
P\left(Gr_{linek\_j}^{4\&T_r}\right) &= \sum_{k\in\{\odot,*,\&\}} P\left(Gr_{linek\_j}^{3\&T_r}|Gr_{linek}^{3\&T_r}\right)P\left(Gr_{linek}^{3\&T_r}\right) \\
&= P\left(Gr_{line\odot\_j}^{4\&T_r}|Gr_{line\odot}^{3\&T_r}\right)P\left(Gr_{line\odot}^{3\&T_r}\right) + P\left(Gr_{line*\_j}^{4\&T_r}|Gr_{line*}^{3\&T_r}\right)P\left(Gr_{line*}^{3\&T_r}\right) + P\left(Gr_{\&\_line\_j}^{3\&T_r}|Gr_{\&\_line}^{3\&T_r}\right)P\left(Gr_{\&\_line}^{3\&T_r}\right) \\
&= \Big\{ \left[P\left(Gr_{linee\_j}^{4\&T_r}|Gr_{linee}^{3\&T_r}\right)P\left(Gr_{linee}^{3\&T_r}|Gr_e^{3\&T_r}\right)P\left(Gr_e^{3\&T_r}|Gr_{alle}^{2\&T_r}\right) + P\left(Gr_{line*\_j}^{4\&T_r}|Gr_{line*}^{3\&T_r}\right)P\left(Gr_{line*}^{3\&T_r}|Gr_*^{3\&T_r}\right)P\left(Gr_{line*}^{3\&T_r}|Gr_{alle}^{2\&T_r}\right) \\
&\quad + P\left(Gr_{\&\_line\_j}^{3\&T_r}|Gr_{\&\_line}^{3\&T_r}\right)P\left(Gr_{\&\_line}^{3\&T_r}|Gr_\&^{3\&T_r}\right)P\left(Gr_{\&\_line}^{3\&T_r}|Gr_{alle}^{2\&T_r}\right)\right] \bullet P\left(Gr_{all\odot}^{2\&T_r}|Gr_{scenic}^{2\&T_r}\right) \bullet P\left(Gr_{scenic}^{2\&T_r}\right)\Big\}
\end{aligned} \tag{8}$$

**(II) Tourist route scale**. The popularity of each tourist route is calculated based on its scenic area, which is jointly contributed to by the popularity of scenic spots ($A^3$, $B^3$ and $C^3$) and tourist routes ($D^3$ and $E^3$). The total popularity of tourist routes is

$$ID\left(Gr_{line}^{3\&T_r}/Gr_{all-line}^{3\&T_r}\right) = P\left(Gr_{line}^{3\&T_r}|Gr_{all-line}^{3\&T_r}\right) = P\left(Gr_{line}^{3\&T_r}\right)/P\left(Gr_{all-line}^{3\&T_r}\right) \tag{9}$$

Among which, $Gr_{line}^{3\&T_r} = Gr_{linee}^{3\&T_r} \bigcup Gr_{line*}^{3\&T_r} \bigcup Gr_{\&\_line}^{3\&T_r} \bigcup Gr_{line1}^{3\&T_r} \bigcup Gr_{\#\_line}^{3\&T_r}$, and $Gr_{all-line}^{3\&T_r} = \bigcup\limits_{line=A}^{X} Gr_{line}^{3\&T_r}$.

1. The popularity of scenic spots is

$$ID\left(\left(\bigcup_{j\in\{linee,line*,\&\_line\}} Gr_j^{3\&T_r}\right)\bigg/ Gr_{all-line}^{3\&T_r}\right) = P\left(\bigcup_{j\in\{line\odot,line*,\&\_line\}} Gr_j^{3\&T_r}\right)\bigg/ P(Gr_{all-line}^{3\&T_r}), \quad (10)$$

which includes

$$A^3 : ID\left(Gr_{linee}^{3\&T_r}/Gr_{all-line}^{3\&T_r}\right) = P\left(Gr_{linee}^{3\&T_r}|Gr_{all-line}^{3\&T_r}\right) = P\left(Gr_{linee}^{3\&T_r}\right)/P\left(Gr_{all-line}^{3\&T_r}\right),$$

$$B^3 : ID\left(Gr_{line*}^{3\&T_r}/Gr_{all-line}^{3\&T_r}\right) = P\left(Gr_{line*}^{3\&T_r}|Gr_{all-line}^{3\&T_r}\right) = P\left(Gr_{line*}^{3\&T_r}\right)/P\left(Gr_{all-line}^{3\&T_r}\right),$$

and

$$C^3 : ID\left(Gr_{\&\_line}^{3\&T_r}/Gr_{all-line}^{3\&T_r}\right) = P\left(Gr_{\&\_line}^{3\&T_r}|Gr_{all-line}^{3\&T_r}\right) = P\left(Gr_{\&\_line}^{3\&T_r}\right)/P\left(Gr_{all-line}^{3\&T_r}\right).$$

2. The popularity of tourist routes is

$$ID\left(\left(\bigcup_{j\in\{line1,\#\_line\}} Gr_j^{3\&T_r}\right)\bigg/ Gr_{all-line}^{3\&T_r}\right) = P\left(\bigcup_{j\in\{line1,\#\_line\}} Gr_j^{3\&T_r}\right)\bigg/ P(Gr_{all-line}^{3\&T_r}), \quad (11)$$

which includes

$$D^3 : ID\left(Gr_{line1}^{3\&T_r}/Gr_{all-line}^{3\&T_r}\right) = P\left(Gr_{line1}^{3\&T_r}|Gr_{all-line}^{3\&T_r}\right) = P\left(Gr_{line1}^{3\&T_r}\right)/P\left(Gr_{all-line}^{3\&T_r}\right),$$

and

$$E^3 : ID\left(Gr_{line\#}^{3\&T_r}/Gr_{all-line}^{3\&T_r}\right) = P\left(Gr_{line\#}^{3\&T_r}|Gr_{all-line}^{3\&T_r}\right) = P\left(Gr_{line\#}^{3\&T_r}\right)/P\left(Gr_{all-line}^{3\&T_r}\right).$$

For example, there are 4 tourist routes, among which the set of scenic spot granules is a subset of the 'scenic spots' at the scenic area scale, and the set of tourist route granules is a subset of the 'tourist routes' at the scenic area scale. Similarly, the same texts are counted for $B^3$, $C^3$ and $E^3$ using the number of actual scenic spots in the text. Taking '*Rizegou*' as an example, the number of $A^3$ is 3,549, and the total number of texts describing tourist routes is 8,858 (that is, the total number of $A^3$, $B^3$, $C^3$, $D^3$ and $E^3$ on the 4 tourist routes). Hence, the popularity of $A^3$ of *Rizegou* is 40.07%.

Some texts may contain multiple scenic spots or multiple tourist routes. We count them repeatedly in multiple tourist route granules, leading to more texts than the actual number of route texts in $Gr_{all-line}^{3\&T_r}$. By considering the distribution of different parts of tourist route granules within a scenic area, the comprehensive popularity of the tourist routes within that scenic

area can be obtained:

$$P\big(Gr_{line}^{3\&T_r}\big) = \sum_{k\in\{e,*,\&,1,\#\}} P\big(Gr_{linek}^{3\&T_r}|Gr_k^{3\&T_r}\big)P\big(Gr_k^{3\&T_r}\big)$$

$$= P\big(Gr_{linee}^{3\&T_r}|Gr_e^{3\&T_r}\big)P\big(Gr_e^{3\&T_r}\big) + P\big(Gr_{line*}^{3\&T_r}|Gr_*^{3\&T_r}\big)P\big(Gr_*^{3\&T_r}\big) + P\big(Gr_{\&\_line}^{3\&T_r}|Gr_\&^{3\&T_r}\big)P\big(Gr_\&^{3\&T_r}\big) + P\big(Gr_{line1}^{3\&T_r}|Gr_1^{3\&T_r}\big)P\big(Gr_1^{3\&T_r}\big)$$

$$+ P\big(Gr_{\#\_line}^{3\&T_r}|Gr_\#^{3\&T_r}\big)P\big(Gr_\#^{3\&T_r}\big) \tag{12}$$

$$= \{[P\big(Gr_{linee}^{3\&T_r}|Gr_e^{3\&T_r}\big)P\big(Gr_e^{3\&T_r}|Gr_{alle}^{2\&T_r}\big) + P\big(Gr_{line*}^{3\&T_r}|Gr_*^{3\&T_r}\big)P\big(Gr_{line*}^{3\&T_r}|Gr_{all\odot}^{2\&T_r}\big) + P\big(Gr_{\&\_line}^{3\&T_r}|Gr_\&^{3\&T_r}\big)P\big(Gr_{\&\_line}^{3\&T_r}|Gr_{alle}^{2\&T_r}\big)]\bullet$$

$$P\big(Gr_{alle}^{2\&T_r}|Gr_{scenic}^{2\&T_r}\big) + [P\big(Gr_{line1}^{3\&T_r}|Gr_1^{3\&T_r}\big)P\big(Gr_1^{3\&T_r}|Gr_{all\#}^{2\&T_r}\big) + P\big(Gr_{\#\_line}^{3\&T_r}|Gr_\#^{3\&T_r}\big)P\big(Gr_\#^{3\&T_r}|Gr_{all\#}^{2\&T_r}\big)] \bullet P\big(Gr_{all\#}^{2\&T_r}|Gr_{scenic}^{2\&T_r}\big)\}P\big(Gr_{scenic}^{2\&T_r}\big).$$

The comprehensive popularity of tourist routes and scenic spots reflects their relative popularity on the macro scale—the relative popularity of scenic areas. The horizontal comparison of popularity between different scenic areas or tourist destinations can be achieved using the comprehensive popularity measures.

**(III) Scenic area scale**. For toponym text, the tourist destination popularity of each scenic area is contributed to by the popularity of scenic spots ($1 - S^{2\&T_r}$, $1 - R\_m - S^{2\&T_r}$, and $m - R\_m - S^{2\&T_r}$($A^2$, $B^2$ and $C^2$ for short)), the popularity of tourist routes ($1 - R^{2\&T_r}$ and $m - R^{2\&T_r}$($D^2$ and $E^2$ for short)), and the popularity of scenic areas.

1. The popularity of scenic spots is

$$ID\big(Gr_{alle}^{2\&T_r}/Gr_{scenic}^{2\&T_r}\big) = P\big(Gr_{alle}^{2\&T_r}|Gr_{scenic}^{2\&T_r}\big) = P\big(Gr_{alle}^{2\&T_r}\big)/P\big(Gr_{scenic}^{2\&T_r}\big), \tag{13}$$

which includes

$$A^2 : ID\big(Gr_e^{3\&T_r}/Gr_{scenic}^{2\&T_r}\big) = P\big(Gr_e^{3\&T_r}|Gr_{scenic}^{2\&T_r}\big) = P\big(Gr_e^{3\&T_r}\big)/P\big(Gr_{scenic}^{2\&T_r}\big),$$

$$B^2 : ID\big(Gr_*^{3\&T_r}/Gr_{scenic}^{2\&T_r}\big) = P\big(Gr_*^{3\&T_r}|Gr_{scenic}^{2\&T_r}\big) = P\big(Gr_*^{3\&T_r}\big)/P\big(Gr_{scenic}^{2\&T_r}\big),$$

and

$$C^2 : ID\big(Gr_\&^{3\&T_r}/Gr_{scenic}^{2\&T_r}\big) = P\big(Gr_\&^{3\&T_r}|Gr_{scenic}^{2\&T_r}\big) = P\big(Gr_\&^{3\&T_r}\big)/P\big(Gr_{scenic}^{2\&T_r}\big).$$

2. The popularity of a tourist route is

$$ID\big(Gr_{all\#}^{2\&T_r}/Gr_{scenic}^{2\&T_r}\big) = P\big(Gr_{all\#}^{2\&T_r}|Gr_{scenic}^{2\&T_r}\big) = P\big(Gr_{all\#}^{2\&T_r}\big)/P\big(Gr_{scenic}^{2\&T_r}\big), \tag{14}$$

which includes

$$D^2 : ID\big(Gr_1^{3\&T_r}/Gr_{scenic}^{2\&T_r}\big) = P\big(Gr_1^{3\&T_r}|Gr_{scenic}^{2\&T_r}\big) = P\big(Gr_1^{3\&T_r}\big)/P\big(Gr_{scenic}^{2\&T_r}\big),$$

and

$$E^2 : ID\big(Gr_\#^{3\&T_r}/Gr_{scenic}^{2\&T_r}\big) = P\big(Gr_\#^{3\&T_r}|Gr_{scenic}^{2\&T_r}\big) = P\big(Gr_\#^{3\&T_r}\big)/P\big(Gr_{scenic}^{2\&T_r}\big).$$

3. The popularity of a scenic area is

$$ID\big(Gr_{scenic1}^{2\&T_r}/Gr_{scenic}^{2\&T_r}\big) = P\big(Gr_{scenic1}^{2\&T_r}|Gr_{scenic}^{2\&T_r}\big) = P\big(Gr_{scenic1}^{2\&T_r}\big)/P\big(Gr_{scenic}^{2\&T_r}\big). \tag{15}$$

For example, the number of toponym texts is 34,290, and the number of texts describing a scenic spot is 8,055 (that is, the sum of $A^2$, $B^2$ and $C^2$). Therefore, the popularity of scenic spots at the scenic area scale is 23.07%.

**(IV) Tourist destination scale**. The popularity of a tourist destination includes the popularity of each scenic area within that tourist destination.

The proportion of toponym text is

$$ID\left(Gr_{scenic}^{2\&T_r}/Gr_{SCENIC}^{2\&T_r}\right) = P\left(Gr_{scenic}^{2\&T_r}|Gr_{SCENIC}^{2\&T_r}\right) = P\left(Gr_{scenic}^{2\&T_r}\right)/P\left(Gr_{SCENIC}^{2\&T_r}\right), \tag{16}$$

and the proportion of nontoponym text is

$$ID\left(\neg Gr_{scenic}^{2\&T_r}/Gr_{SCENIC}^{2\&T_r}\right) = P\left(\neg Gr_{scenic}^{2\&T_r}|Gr_{SCENIC}^{2\&T_r}\right) = P\left(\neg Gr_{scenic}^{2\&T_r}\right)/P\left(Gr_{SCENIC}^{2\&T_r}\right) \tag{17}$$

For example, the total number of texts describing *Jiuzhaigou* is 36,740, the number of toponym texts is 34,290; thus, the proportion of toponym texts is 93.33%.

**4.2.2 Tourist destination popularity at different temporal scales.** The tourism text dataset organized based on the temporal dimension can also support TDP calculations at different temporal scales [79]. Combined with the spatial dimension, the temporal dimension is helpful for exploring the temporal features and evolutionary rules of tourist group behaviors.

1. The year scale is

$$ID\left(\neg Gr_j^{S_r\&1-a}/Gr_j^{S_r\&1}\right) = P\left(\neg Gr_j^{S_r\&1-a}|Gr_j^{S_r\&1}\right) = P\left(\neg Gr_j^{S_r1-a}\right)/\sum_{a\in YearSet} P\left(Gr_j^{S_r\&1-a}\right), \tag{18}$$

2. the month scale is

$$ID\left(Gr_j^{S_r\&2-a/b}/Gr_j^{S_r\&1-a}\right) = P\left(Gr_j^{S_r\&2-a/b}|Gr_j^{S_r\&1-a}\right) \bullet P\left(Gr_j^{S_r\&1-a}|Gr_j^{S_r\&1}\right) = \frac{P\left(Gr_j^{S_r\&2-a/b}\right)}{\sum\limits_{b\in MonthSet} P\left(Gr_j^{S_r\&2-a/b}\right)} \bullet \frac{P\left(Gr_j^{S_r\&1-a}\right)}{\sum\limits_{a\in YearSet} P\left(Gr_j^{S_r\&1-a}\right)}, \tag{19}$$

3. the day scale is

$$ID\left(Gr_j^{S_r\&3-a/b/c}/Gr_j^{S_r\&2-a/b}\right) = P\left(Gr_j^{S_r\&3-a/b/c}|Gr_j^{S_r\&2-a/b}\right) \bullet P\left(Gr_j^{S_r\&2-a/b}|Gr_j^{S_r\&1-a}\right) = \frac{P\left(Gr_j^{S_r\&3-a/b/c}\right)}{\sum\limits_{c\in DaySet} P\left(Gr_j^{S_r\&3-a/b/c}\right)} \bullet \frac{P\left(Gr_j^{S_r\&2-a/b}\right)}{\sum\limits_{b\in MonthSet} P\left(Gr_j^{S_r\&2-a/b}\right)} \tag{20}$$

$$ID\left(Gr_j^{S_r\&3-a/b/c}/Gr_j^{S_r\&1-a}\right) = P\left(Gr_j^{S_r\&3-a/b/c}|Gr_j^{S_r\&1-a}\right) \bullet P\left(Gr_j^{S_r\&1-a}|Gr_j^{S_r\&1}\right) = \frac{\sum\limits_{b\in MonthSet} P\left(Gr_j^{S_r\&1-a/b/c}\right)}{\sum\limits_{c\in DaySet}\sum\limits_{b\in MonthSet} P\left(Gr_j^{S_r\&1-a/b/c}\right)} \bullet \frac{P\left(Gr_j^{S_r\&1-a}\right)}{\sum\limits_{a\in YearSet} P\left(Gr_j^{S_r\&1-a}\right)}, \tag{21}$$

4. and the hour scale is

$$ID\left(Gr_j^{S_r\&4-a/b/c/d}/Gr_j^{S_r\&3-a/b/c}\right) = P\left(Gr_j^{S_r\&4-a/b/c/d}|Gr_j^{S_r\&3-a/b/c}\right) \bullet P\left(Gr_j^{S_r\&3-a/b/c}|Gr_j^{S_r\&2-a/b}\right)$$

$$= \frac{P\left(Gr_j^{S_r\&4-a/b/c/d}\right)}{\sum\limits_{d\in HourSet} P\left(Gr_j^{S_r\&4-a/b/c/d}\right)} \bullet \frac{P\left(Gr_j^{S_r\&3-a/b/c}\right)}{\sum\limits_{c\in DaySet} P\left(Gr_j^{S_r\&3-a/b/c}\right)} \tag{22}$$

$$ID\left(Gr_j^{S_r\&4-a/b/c/d}/Gr_j^{S_r\&2-a/b}\right) = P\left(Gr_j^{S_r\&4-a/b/c/d}|Gr_j^{S_r\&3-a/b/c}\right) \bullet P\left(Gr_j^{S_r\&3-a/b/c}|Gr_j^{S_r\&2-a/b}\right)$$

$$= \frac{\sum\limits_{c\in DaySet} P\left(Gr_j^{S_r\&4-a/b/c/d}\right)}{\sum\limits_{b\in MonthSet}\sum\limits_{c\in DaySet} P\left(Gr_j^{S_r\&4-a/b/c/d}\right)} \bullet \frac{P\left(Gr_j^{S_r\&2-a/b}\right)}{\sum\limits_{b\in MonthSet} P\left(Gr_j^{S_r\&2-a/b}\right)} \tag{23}$$

$$ID\left(Gr_j^{S_r\&4-a/b/c/d}/Gr_j^{S_r\&1-a}\right) = P\left(Gr_j^{S_r\&1-a/b/c/d}|Gr_j^{S_r\&1-a/b/c}\right) \bullet P(Gr_j^{S_r\&1-a}|Gr_j^{S_r\&1})$$

$$= \frac{\sum\limits_{b\in MonthSet}\sum\limits_{c\in DaySet} P\left(Gr_j^{S_r\&4-a/b/c/d}\right)}{\sum\limits_{d\in HourSet}\left(\sum\limits_{b\in MonthSet}\sum\limits_{c\in DaySet} P\left(Gr_j^{S_r\&4-a/b/c/d}\right)\right)} \bullet \frac{P(Gr_j^{S_r\&1-a})}{\sum\limits_{a\in YearSet} P(Gr_j^{S_r\&1-a})}. \tag{24}$$

# 5 A case study from *Jiuzhaigou*

## 5.1 Descriptions of the research area and data

*Jiuzhaigou* is a famous national scenic area, a nature reserve, and a typical waterscape and landscape scenic area in China [80–81] that spans a large area, features beautiful scenery, and attracts numerous tourists. The *Sina* microblog site is rich in related scenic data and is highly representative.

Our research group purchased the commercial *Sina* microblog application programming interface (API) and downloaded the microblog data within the spatial range of the scenic area. The microblog data used in this study was collected from 00:00:00 on January 1, 2013 to 00:00:00 on January 1, 2018. *Sina* microblog provides the data collection method of "collection point" and "range" to cover the spatial range of the research area. The followings are the parameters used in the experiment:

Collection point 1: latitude 33.216981, longitude 103.912572, range: 10000m;

Collection point 2: latitude 33.079005, longitude 103.898839, range: 10000m.

Finally, the data within the coverage area are filtered according to the method mentioned in the manuscript.

In total, we collected 105,226 microblog posts from 2013 to 2017, which constitutes all the *Sina* microblog posts published during this period regarding *Jiuzhaigou*. By filtering noise data (the number of noise data is 68,486), we obtained 36,740 valid tourism text entries (see Table 1) that constitute the dataset.

**Table 1. Effective microblogs in *Jiuzhaigou* in 2013–2017.**

| Data | Year | Number of valid texts | Description of main attributes |
|---|---|---|---|
| *Jiuzhaigou* scenic area | 2013 | 9,431 | UID: user ID; |
| | 2014 | 6,339 | Created_at: release time; |
| | 2015 | 6,179 | Lat/Long: Release position; |
| | 2016 | 7,514 | Text: text content; |
| | 2017 | 7,277 | User name: user name. |
| **Total** | | 36,740 | |

## 5.2 Tourist destination popularity mining at multi-spatiotemporal scales

TDPMTGC integrates spatial and temporal scales into one systematic model by using GrC, which makes all the scales in spatial and temporal dimensions related. For each spatiotemporal scale, not only can the popularity of spatiotemporal units represented by each data granule be quantitatively described but the contributions of different units to the overall TDP of this scale can also be compared. The popularity contribution of data granules in the lower layer to those in the upper layer can be further quantitatively analyzed. Due to the length restrictions of this paper, we take only the correlation shown in Fig 3 as an example to conduct multi-spatiotemporal scale TDP mining to demonstrate the superior performance of TDPMTGC. The results are shown in Tables 2–5 and Figs 4 and 5.

**5.2.1 The spatiotemporal model associated with the scenic area.** **General characteristics**: at the scenic area scale, TDP mainly includes the *Jiuzhaigou* features of the scenic area and the features of each tourist route (see Table 3). The contribution of *Jiuzhaigou* in the scenic area reaches 76%, while those from different routes or scenic spots account for only

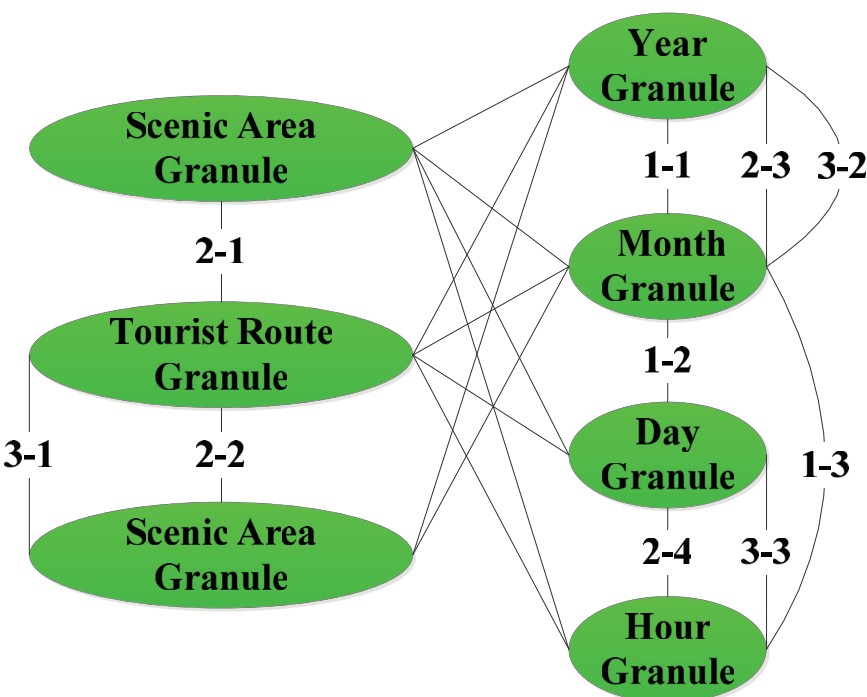

**Fig 3. Correlation diagram of multi-spatiotemporal scale data in *Jiuzhaigou*.**

**Table 2. Calculated results of TDP (abbreviated pop.) at *Jiuzhaigou* tourist destination scales.**

| *Jiuzhaigou* Scenic Area | | | | Scenic Area 2 | Scenic Area... | Scenic Area N | Total text of *Jiuzhaigou* |
|---|---|---|---|---|---|---|---|
| Toponym text | Proportion | Nontoponym text | Proportion | | | | |
| 34290 | 93.33% | 2450 | 6.67% | omit | omit | omit | 36740 |

**Table 3. Calculated results of TDP (abbreviated pop.) at *Jiuzhaigou* scenic area scales.**

| Pop. of scenic spots | | | | | | Pop. of tourist routes | | | | Pop. of scenic areas | | Number of toponym text |
|---|---|---|---|---|---|---|---|---|---|---|---|---|
| $1\text{-}S^{2\&Tr}$ | Pop. | $1\text{-}R\_m\text{-}S^{2\&Tr}$ | Pop. | $m\text{-}R\_m\text{-}S^{2\&Tr}$ | Pop. | $1\text{-}R^{2\&Tr}$ | Pop. | $m\text{-}R^{2\&Tr}$ | Pop. | Scenic Area | Pop. | |
| 6717 | 19.59% | 739 | 2.16% | 599 | 1.75% | 72 | 0.21% | 15 | 0.04% | 26148 | 76.26% | 34290 |

**Table 4. Calculated results of TDP (abbreviated pop.) at *Jiuzhaigou* tourist route scales.**

| Pop. of scenic spots | | | | | | Pop. of tourist routes | | | | Total of tourist routes | Sum of pop. | Comprehensive pop. |
|---|---|---|---|---|---|---|---|---|---|---|---|---|
| $1\text{-}S^{3\&Tr}$ | Pop. | $1\text{-}R\_m\text{-}S^{3\&Tr}$ | Pop. | $m\text{-}R\_m\text{-}S^{3\&Tr}$ | Pop. | $1\text{-}R^{3\&Tr}$ | Pop. | $m\text{-}R^{3\&Tr}$ | Pop. | | | |
| 903 | 10.19% | 109 | 1.23% | 293 | 3.31% | 24 | 0.27% | 9 | 0.10% | 1338 | 15.10% | 3.90% |
| 3549 | 40.07% | 419 | 4.73% | 533 | 6.02% | 35 | 0.40% | 13 | 0.15% | 4549 | 51.35% | 13.27% |
| 2245 | 25.34% | 209 | 2.36% | 464 | 5.24% | 13 | 0.15% | 11 | 0.12% | 2942 | 33.21% | 8.58% |
| 20 | 0.23% | 2 | 0.02% | 7 | 0.08% | 0 | 0 | 0 | 0 | 29 | 0.33% | 0.08% |

**Table 5. Calculated results of TDP (abbreviated pop.) at *Jiuzhaigou* scenic spot scales.**

| Name | | Type | Pop. of scenic spots | | | | | | | | Comprehensive Pop. |
|---|---|---|---|---|---|---|---|---|---|---|---|
| | | | $1\text{-}S^{4\&Tr}$ | Pop. | $1\text{-}R\_m\text{-}S^{4\&Tr}$ | Pop. | $m\text{-}R\_m\text{-}S^{4\&Tr}$ | Pop. | Total of scenic spots | Sum of Pop. | |
| *Shuzhenggou* | *Heyezhai* | village | 19 | 0.18% | 1 | 0.01% | 9 | 0.08% | 29 | 0.27% | 0.08% |
| | *Yanazhai* | village | 0 | 0 | 0 | 0 | 0 | 0 | 0 | 0 | 0 |
| | *Panyazhai* | village | 0 | 0 | 0 | 0 | 0 | 0 | 0 | 0 | 0 |
| | *Jianpanzhai* | village | 0 | 0 | 0 | 0 | 0 | 0 | 0 | 0 | 0 |
| | *Guwazhai* | village | 0 | 0 | 0 | 0 | 0 | 0 | 0 | 0 | 0 |
| | *Penjingtan* | beach | 28 | 0.26% | 11 | 0.10% | 21 | 0.20% | 60 | 0.57% | 0.17% |
| | *Luweihai* | lake | 135 | 1.27% | 21 | 0.20% | 64 | 0.60% | 220 | 2.07% | 0.64% |
| | *Heijiaozhai* | village | 0 | 0 | 0 | 0 | 1 | 0.01% | 1 | 0.01% | 0 |
| | *Zhayizhagashenshan* | mountain | 2 | 0.02% | 0 | 0 | 1 | 0.01% | 3 | 0.03% | 0.01% |
| | *Shuanglonghai* | lake | 14 | 0.13% | 15 | 0.14% | 15 | 0.14% | 44 | 0.41% | 0.13% |
| | *Huohuahai* | lake | 119 | 1.12% | 31 | 0.29% | 69 | 0.65% | 219 | 2.06% | 0.64% |
| | *Huohuahaipubu* | waterfall | 0 | 0 | 0 | 0 | 0 | 0 | 0 | 0 | 0 |
| | *Wolonghai* | lake | 11 | 0.10% | 11 | 0.10% | 15 | 0.14% | 37 | 0.35% | 0.11% |
| | *Shuzhengqunhai* | lake | 67 | 0.63% | 26 | 0.25% | 38 | 0.36% | 131 | 1.24% | 0.38% |
| | *Shuzhengpubu* | waterfall | 109 | 1.03% | 30 | 0.28% | 65 | 0.61% | 204 | 1.92% | 0.59% |
| | *Shuzhengzhai* | village | 148 | 1.40% | 24 | 0.23% | 41 | 0.39% | 213 | 2.01% | 0.62% |
| | *Laohuhai* | lake | 118 | 1.11% | 47 | 0.44% | 81 | 0.76% | 246 | 2.32% | 0.72% |
| | *Xiniuhai* | lake | 133 | 1.25% | 46 | 0.43% | 119 | 1.12% | 298 | 2.81% | 0.87% |

*(Continued)*

**Table 5.** (Continued)

| Name | | Type | Pop. of scenic spots | | | | | | | | Comprehensive Pop. |
|---|---|---|---|---|---|---|---|---|---|---|---|
| | | | $1\text{-}S^{4\&Tr}$ | Pop. | $1\text{-}R\_m\text{-}S^{4\&Tr}$ | Pop. | $m\text{-}R\_m\text{-}S^{4\&Tr}$ | Pop. | Total of scenic spots | Sum of Pop. | |
| Rizegou | Nuorilangqunhai | lake | 0 | 0 | 1 | 0.01% | 0 | 0 | 1 | 0.01% | 0.00% |
| | **Nuorilangpubu** | **waterfall** | **389** | **3.67%** | **52** | **0.49%** | **165** | **1.56%** | **606** | **5.71%** | **1.77%** |
| | Nuorilangbaohuzhongxin | other | 0 | 0 | 0 | 0 | 0 | 0 | 0 | 0 | 0 |
| | Semonvshenshan | mountain | 0 | 0 | 0 | 0 | 2 | 0.02% | 2 | 0.02% | 0.01% |
| | **Jinghai** | **lake** | **301** | **2.84%** | **85** | **0.80%** | **97** | **0.91%** | **483** | **4.55%** | **1.41%** |
| | Dagenanshenshan | mountain | 0 | 0 | 0 | 0 | 0 | 0 | 0 | 0 | 0 |
| | **Zhenzhutan** | **beach** | **215** | **2.03%** | **85** | **0.80%** | **79** | **0.74%** | **379** | **3.57%** | **1.11%** |
| | **Zhenzhutanpubu** | **waterfall** | **449** | **4.23%** | **89** | **0.84%** | **91** | **0.86%** | **629** | **5.93%** | **1.83%** |
| | Jinlinghai | lake | 2 | 0.02% | 3 | 0.03% | 2 | 0.02% | 7 | 0.07% | 0.02% |
| | Kongquehedao | lake | 3 | 0.03% | 10 | 0.09% | 4 | 0.04% | 17 | 0.16% | 0.05% |
| | **Wuhuahai** | **lake** | **931** | **8.78%** | **211** | **1.99%** | **246** | **2.32%** | **1388** | **13.09%** | **4.05%** |
| | **Xiongmaohai** | **lake** | **299** | **2.82%** | **160** | **1.51%** | **133** | **1.25%** | **592** | **5.58%** | **1.73%** |
| | Xiongmaohaipubu | waterfall | 20 | 0.19% | 22 | 0.21% | 5 | 0.05% | 47 | 0.44% | 0.14% |
| | **Jianzhuhai** | **lake** | **359** | **3.38%** | **138** | **1.30%** | **113** | **1.07%** | **610** | **5.75%** | **1.78%** |
| | Jianzhuhaipubu | waterfall | 95 | 0.90% | 25 | 0.24% | 18 | 0.17% | 138 | 1.30% | 0.40% |
| | Rizegoubaohuzhongxin | other | 0 | 0 | 0 | 0 | 0 | 0 | 0 | 0 | 0 |
| | Tian'ehai | lake | 25 | 0.24% | 25 | 0.24% | 6 | 0.06% | 56 | 0.53% | 0.16% |
| | Fangcaohai | lake | 4 | 0.04% | 12 | 0.11% | 3 | 0.03% | 19 | 0.18% | 0.06% |
| | Jianyanxuanquan | waterfall | 0 | 0 | 1 | 0.01% | 0 | 0 | 1 | 0.01% | 0 |
| | **Yuanshisenlin** | **forest** | **457** | **4.31%** | **55** | **0.52%** | **73** | **0.69%** | **585** | **5.52%** | **1.71%** |
| | Zangmalonglihai | lake | 0 | 0 | 0 | 0 | 0 | 0 | 0 | 0 | 0 |
| Zechawagou | Zechawazhai | village | 12 | 0.11% | 0 | 0 | 8 | 0.08% | 20 | 0.19% | 0.06% |
| | Xiajijiehai | lake | 3 | 0.03% | 10 | 0.09% | 3 | 0.03% | 16 | 0.15% | 0.05% |
| | Ganzigonggaishan | mountain | 0 | 0 | 0 | 0 | 0 | 0 | 0 | 0 | 0 |
| | Shangjijiehai | lake | 6 | 0.06% | 4 | 0.04% | 2 | 0.02% | 12 | 0.11% | 0.03% |
| | **Wucaichi** | **lake** | **1456** | **13.73%** | **205** | **1.93%** | **345** | **3.25%** | **2006** | **18.91%** | **5.85%** |
| | **Changhai** | **lake** | **768** | **7.24%** | **201** | **1.89%** | **288** | **2.72%** | **1257** | **11.85%** | **3.67%** |
| Zharugou | Zharusi | temple | 19 | 0.18% | 2 | 0.02% | 4 | 0.04% | 25 | 0.24% | 0.07% |
| | Baojingyan | mountain | 1 | 0.01% | 2 | 0.02% | 3 | 0.03% | 6 | 0.06% | 0.02% |
| | Rexizhai | village | 0 | 0 | 0 | 0 | 0 | 0 | 0 | 0 | 0 |
| | Guoduzhai | village | 0 | 0 | 0 | 0 | 0 | 0 | 0 | 0 | 0 |

approximately 24%. More than 90% of the texts reference a single tourist route (Single-Spot, 1-Route-N-spots and Single-Route). Overall, most descriptions of *Jiuzhaigou* describe the entire scenic area.

**Correlation 1–1 (scenic area, year-month)**: the monthly changing patterns are basically the same across 2014, 2015 and 2016. The peak-season lasted from June to October. The popularity began to increase significantly in June, reaching a small peak in August, slightly decreasing in September, and then reaching the annual peak in October, while the off-season lasted from November to May of the following year. The popularity decreased significantly in November and reached its lowest trough in December and January before steadily recovering from February to May. These observations are consistent with the conclusions reached by Wang [5] and Yan [82]. However, significant abnormal trends occurred in 2013 and 2017. The two typical peaks present in June and October of 2013 were much higher than those in other

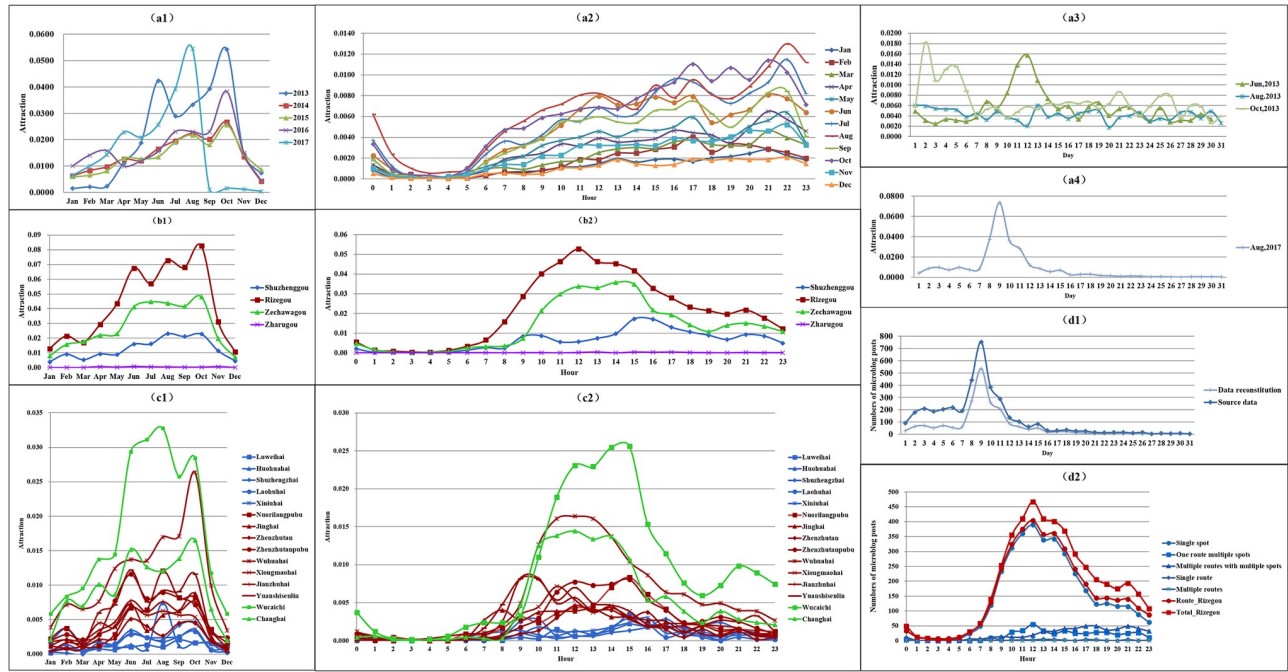

**Fig 4. The variation tendencies of the popularity of *Jiuzhaigou* at various temporal scales.** (a1) scenic areas at the monthly scale $L^{1\&2}$. (a2) scenic areas at the hourly scale $L^{1\&4}$. (a3) scenic areas for 2013 at the daily scale $L^{1\&3\_2013/b/c}$. (a4) scenic areas for 2017 at the daily scale $L^{1\&3\_2017/b/c}$. (b1) tourist routes at the monthly scale $L^{1\&2}$. (b2) tourist routes at the hourly scale $L^{1\&4}$. (c1) scenic spots at the monthly scale $L^{1\&2}$. (c2) scenic spots at the hourly scale $L^{1\&4}$. (d1) popularity based on the numbers of microblog posts in August 2017. (d2) popularity based on the numbers of microblog posts in *Rizegou*.

years. A sudden increase in popularity also occurred in August 2017 and then sharply decreased after reaching the peak. These anomalies are related to policy or tourism events and can be further interpreted and analyzed based on the distribution patterns at the daily scale.

**Correlation 1–2 (scenic area, month-day)**: At the daily scale, we selected several months for which to analyze the daily popularity changes in 2013 and 2017. The variations in daily popularity in June, August and October 2013 are illustrated in Fig 4(b1). The daily variation in August fluctuates randomly. The abnormal popularity in June 2013 occurred mainly from the 10th to 13th, which overlapped with the Dragon Boat Festival holiday, i.e., the second holiday in which the free expressway was implemented in October 2012, thus intensifying tourists' desire to travel and leading to a sudden increase in TDP. The abnormal popularity in October lasted mainly from the 2nd to 6th, coinciding with the National Day holiday. The popularity reached its highest value on October 2, corresponding to the large-scale tourist detention event on that day. The daily variation in the August anomaly in 2017 had its highest peak from August 8–11 as shown in Fig 4(b2), which coincided with the period of the 7.0-magnitude earthquake in *Jiuzhaigou* County on August 8. These results are consistent with the conclusions of Cao [83], which indicated that the disaster event was the main factor leading to the increase in popularity in *Jiuzhaigou* during this period.

**Correlation 1–3 (scenic area, month-hour)**: The popularity continuously increases from 6:00 to 22:00, with a slightly fluctuating pattern. In addition, the daily variation in hot months is significantly different from those in other months. During the peak popularity months from June to October, the daily variation shows a pattern with three peaks and two valleys: a peak at approximately 12:00, a higher peak at 15:00–16:00, the highest peak at 21:00–22:00, a valley at 13:00–14:00 and the lowest trough at 18:00–19:00. This pattern is consistent with the

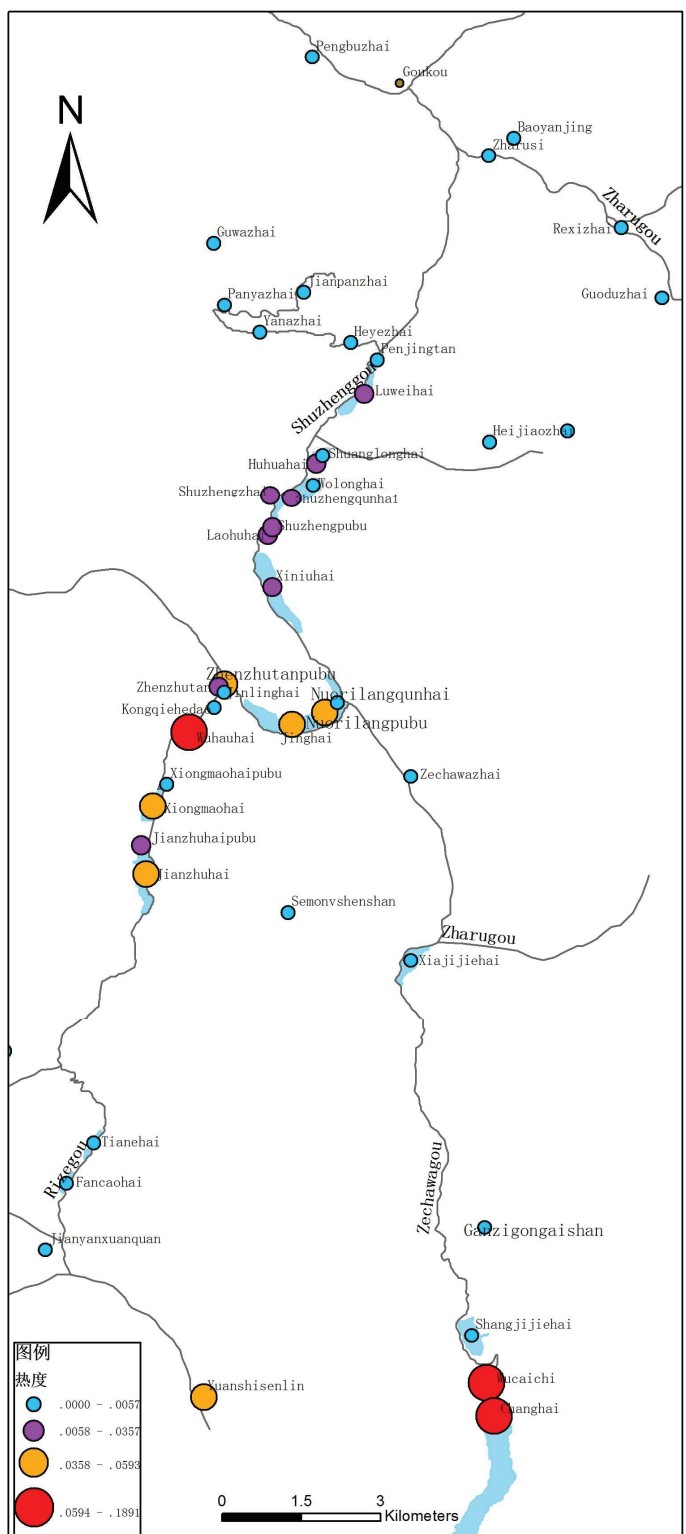

**Fig 5. Comprehensive popularity of scenic spots represented by text data granules at the scenic spot scale.**

characteristics of sightseeing tours, dining, and rest times in summer and autumn. During the off-season months from November to May, the daily variation shows a pattern of two peaks and one valley (peaks at 16:00–17:00 and 21:00–22:00 and a valley at 18:00–19:00), or an insignificant peak and valley. These patterns are associated with the characteristics of sightseeing tours, dining, and rest times in winter and spring.

**5.2.2 The spatiotemporal model associated with tourist route.   Correlation 2–1 (tourist route-scenic area)**: The popularity rankings of the four tourist routes from high to low are *Rizegou*, *Zechawagou*, *Shuzhenggou*, and *Zharugou*. *Rizegou* contains the most scenic spots, and the popularity of this tourist route contributes >50% of the popularity of all tourism routes (see Table 4). With 6 representative landscapes, the popularity of *Zechawagou* reached 33%, ranking second among the four routes. Although many *Shuzhenggou* contains many scenic spots, its popularity is lower than those of *Rizegou* and *Zechawagou*. Few tourists pay attention to *Zharugou*, resulting in very low popularity value.

**Correlation 2–2 (tourist route—scenic spot)**: The popularity of each tourist route is mainly contributed to by the scenic spots on each route, whose contribution rate is close to 100%, among which the contribution rates of single scenic spots are all >67%. Microblog users who mention tourist route spatial units rarely describe the names of tourism routes but they do directly describe specific scenic spots, and most describe single scenic spots.

**Correlation 2–3 (tourist route-scenic spot, year-month)**: The popularity variation tendencies of tourist routes at the monthly scale within a year were basically consistent with those of the scenic area scale. The high popularity period lasted from June to October. The high popularity of *Rizegou* shows an obvious pattern of three peaks and two valleys, which is consistent with the high popularity of scenic area in June and October, indicating the contribution of *Rizegou*'s popularity to its scenic area. There is no inter-monthly fluctuation based on the popularity anomalies in each route, which means no popularity anomaly was caused by inter-monthly or seasonal landscapes with higher popularity.

**Correlation 2–3 (tourist route-scenic spot)**: There were significant differences in the variation tendency of the three main lines within the day. ① The popularity of *Rizegou* is always the highest, while *Shuzhenggou* is always the lowest, indicating that visitors pay different attention to those routes. ② The hours during which the popularity of *Rizegou*, *Zechawagou* and *Shuzhenggou* increased significantly were 8:00, 9:00 and 15:00, respectively; the peak hours are 10:00–15:00, 11:00–15:00 and 15:00–17:00, respectively, and the peaks occurred at 12:00, 14:00 and 15:00, respectively, while the troughs appeared at 20:00, 19:00 and 20:00, respectively. The above characteristics indicate that the route popularity is obviously affected by tourism guides, and most tourists start by entering *Rizegou* and *Zechawagou* and only later visit *Shuzhenggou*. The popularity variation tendency of the route can be further analyzed by the scenic spot scale model.

**5.2.3 The spatiotemporal model associated with scenic spots.   **At the scenic spot scale, the popularity of the top-15 hot spots accounts for approximately 90% of all the scenic spots; therefore, we selected only the Top-15 scenic spots for this discussion.

**Correlation 3–1 (scenic spot-tourist route)**: the popularity of scenic spots with different types or on different tourist routes differ (see Table 5 and Fig 5). Three scenic spots in the first level have the highest popularity: *Wucaichi* and *Changhai* in *Zechawagou* and *Wuhuahai* in *Rizegou* (in bold underlined font). The scenic spots in the second level with high popularity are concentrated in *Rizegou*, including the 7 scenic spots of *Zhenzhutanpubu*, *Jianzhuhai*, *Nuorilangpubu*, *Xiongmaohai*, *Yuanshisenlin*, *Jinghai* and *Zhenzhutan* (in bold font). The scenic spots in *Shuzhenggou* are ranked only at the third level and include *Luweihai*, *Huohuahai*, *Shuzhengzhai*, *Laohuhai* and *Xiniuhai* (underlined font). In general, most of the popular scenic spots are water-related landscapes, such as lakes and waterfalls; these account for

approximately 1/3 of the scenic spots, while the other scenic spots have relatively lower popularity, among which folk customs and cultural landscapes (such as villages) have the lowest popularity. These results are highly consistent with the conclusion of Tang [40] that 'most tourists consider the natural landscape in *Jiuzhaigou*, while the Tibetan villages and other cultural landscapes with important folk culture and historical values are not recognized enough'. At the same time, TDPMTGC quantitatively analyzed the contribution of granules in the lower layer to the TDP of the upper layers, improving the understanding of tourist behaviors.

The type characteristics of scenic spot popularity can explain the differences in route popularity from another perspective. The lakes of *Rizegou* account for approximately half of the route, and the 4 hot lakes effectively improve the popularity of the route. There are fewer lakes in *Zecha*wagou, but the first-level-popularity landscapes *Wucaichi* and *Changhai* support the high route popularity. There are more scenic spots in *Shuzhenggou*, but half of them are villages, which have low popularity, leading to the low route popularity.

**Correlation 3–2 (scenic spot-tourist route, year-month)**: the variation tendency of scenic spot popularity at the monthly scale shows three modes: single peak, double peak and triple peak. ① Single peak: The single peak of *Wuhuahai*, *Zhenzhutanpubu*, *Xiongmaohai*, *Nuorilangpubu* in *Rizegou* appears in June, August and October, while the peak of *Huohuahai* in *Shuzhenggou* appears in August. ② Double peak: the double peak of *Rizegou* appears in June and October and includes *Yuanshisenlin*, *Jinghai* and *Zhenzhutan*. The double peak of *Zechawagou*, namely *Wucaichi* and *Changhai*, appears in June, August and October. The double peak of *Shuzhenggou* appears in June and October and includes scenic spots such as *Shuzhengzhai*, *Laohuhai* and *Xiniuhai*. ③ Triple peak: The triple peak appears only for *Jianzhuhai* in *Rizegou*, also in June, August and October. By analyzing the proportions of the peak months of each scenic spot in the annual popularity of the scenic spot and their contribution to the annual popularity of tourist routes, the characteristics of the popularity of scenic spots on tourist routes can be obtained. For example, *Zhechawagou* has two of the highest-popularity scenic spots: *Wucaichi* (double peak in August and October) and *Changhai* (double peak in June and October). The peak months accounted for <23.89%, 20.68%>, <20.99%, 22.79%> of the annual popularity of scenic spots, while their contributions to the annual popularity of tourist routes are <10.51%, 9.1%>, <4.87%, 5.29%>, respectively. Therefore, the tourist route peaks in October and the peak season lasts from June to October. In summary, the peaks of high-popularity scenic spots all appeared in June, August and October, and the total numbers of peaks are 10, 4 and 10, respectively. The proportion of each monthly peak in the annual popularity of scenic spots accounts for 20%–33% (*Huohuahai* reaches approximately 66%); the contributions of the first-level popular scenic spots to the popularity of tourist routes ranges from 4.87–10.51%, and the contribution of the second-level and third-level popular scenic spots to the popularity of tourist routes is generally less than 3%. The popularity of a tourist route is affected by its scenic spots, and the appearance and duration of its peaks are consistent with those of its scenic spots. Combined with the monthly variation tendency of scenic spots, the tourist route and scenic area popularity modes are relatively consistent.

**Correlation 3–3 (scenic spot-tourist route, day-hour)**: The initial period, peak type and popularity level of its scenic spots have a significant influence on the popularity mode of the tourist route. For example, the popularity of scenic spots in *Rizego*u begins to rise at 7:00–9:00, and the time spans of their peak occurrences are large. The peak of *Wuhuahai* at 12:00 makes a large contribution to its tourist route. The peak values in *Yuanshisenlin* and *Jianzhuhai* reach 0.82% at 9:00 and last for 2 hours. The peak values of *Xiongmaohai*, *Wuhuahai* and *Jinghai* are 0.71%, 1.64% and 0.42%, respectively, from 11:00–13:00, lasting for 1~3 hours. The peak values of *Zhenzhutanpubu* and *Nuorilangpubu* are 0.008 at 15:00, lasting for 1~3 hours. After accumulation, the popularity of *Rizegou* rises at 7:00, its peak appears at 12:00, and its high popularity

lasts from 11:00–15:00 with a popularity value ranging from 3.83%–4.39%. Similarly, through the superposition of the popularity of all scenic spots in their tourist route, the popularity of *Zechawagou* increases at 9:00 and reaches a double peak with a popularity value between 2.51% and 2.97% from 11:00–15:00, while the popularity of *Shuzhenggou* presents a pattern rises slightly and then falls at 8:00. Its popularity rises again significantly at 14:00, a peak appears at 15:00, and this peak lasts from 15:00–16:00 with a popularity value from 1.24%–1.4%.

**5.2.4 The relationship between popularity variation tendency and numbers of micro-blog posts.** The temporal variation of the TDP is calculated based on the numbers of micro-blog posts during the same time period. The two variations are similar but not identical, and there are three main differences.

1. Source data and reorganized data. The temporal variations in TDP as calculated by TDPMTGC are based on the text dataset after data reorganization rather than on the source data of microblog posts during the same period of the research area. Taking the data in 2017 as an example, 20,764 pieces of source data were focused on *Jiuzhaigou* in 2017, although this number was reduced to 7,277 after data reorganization. A comparison of the daily variation patterns within months (see Fig 4(d1)) showed that their overall trend was consistent and both were affected by the earthquake in *Jiuzhaigou* on August 8. However, the source data contain texts that are unrelated to the research area; thus, the variations are not exactly the same.

2. Intersections between data granules. Intersections occur between data granules at some spatial scales, and the intersecting parts of the text belong to multiple granules. When calculating the comprehensive popularity of data granules, it is necessary to include the intersecting parts of the text in multiple granules at the same time, resulting in a text expansion compared with the source data, and these variations are slightly different from the changing trends in the number of microblog posts. For example, the route granules at the tourist route scale include a single spot, one route with multiple spots, multiple routes with multiple spots, single route and multiple routes, among which multiple routes with multiple spots and multiple routes granules simultaneously belong to multiple route granules. Therefore, the absolute number of routes is slightly different from the overall number of micro-blog posts (see Fig 4(d2)).

3. The popularity value of the same data granules can be different at different scales. For example, the popularity of *Wucaichi* at the scenic spot scale is 18.91% as calculated based on scenic spots, while its comprehensive popularity in the scenic area is 5.85% (see Table 5). Moreover, due to the different inclusion relationships of data granules at different scales, there is not necessarily a proportional relationship between the popularity values (i.e., multiple routes with multiple spots granules belong to multiple tourist route granules at the tourist route scale but only to one granule in the scenic area scale and thus are calculated differently on different scales).

**5.2.5 Summary.** TDPMTGC allows for the comparability of the data granules of all spatial-spatial, temporal-temporal and spatial-temporal layers and then achieves the comparison of popularity values of data granules between adjacent scales and across scales. Detailed and quantitative descriptions of TDP at multi-spatiotemporal scales are helpful for comprehensively and deeply exploring the spatiotemporal characteristics of tourism from the viewpoint of tourists' cognition.

## 6 Discussion

Guided by the data granulation approach, TDPMTGC normalizes and recombines the spatiotemporal information expressed explicitly or implicitly in unstructured tourism texts into a systematic framework that integrates the spatial and temporal dimensions, ensuring that its popularity calculations are measurable and comparable.

Previous TDP research approaches have important implications for this paper. We take the approaches of Hu [41], Wang [5], Tang [40] as examples and compare them with the TDPMTGC proposed in this paper to both acknowledge the inheritance TDPMTGC owes to the existing approaches as well as its further innovations and to reflect its potential advantages and value in future applications, leading to further research questions.

1. Dataset. Before the advent of the big data era, questionnaires represented the main method of obtaining user data (e.g., Tang et al [40]). However, the rapid development of the Internet has caused the data scale to explode. Increasingly, scholars focus on mining social media data, such as Flickr photos and microblog data (e.g., Hu et al [41] and Wang et al [5]). TDPMTGC uses the full content of tourism UGC texts, which contain rich spatiotemporal and semantic information that is conducive to in-depth explorations of the rules governing tourists' spatiotemporal behaviors and analysis of the driving mechanisms of tourism spatial patterns and processes. This approach better reflects users' real emotional trends than does data collected based on specific research objectives, such as questionnaire surveys and interviews, and it reduces the differences caused by sparse or inconsistent samples. For example, analyzing the variation tendency of popularity of *Jiuzhaigou* at the daily scale, we find that the unusual period of attention by tourists is associated with holidays, special policies, tourism events and sudden disasters. The feature extraction of tourism UGC text from an abnormal time period can be used to analyze users' emotional trends. One advantage of TDPMTGC is that the data types it can use are unrestricted. Although we chose text for this study, other types of data could also be employed, and we plan to conduct further research using Flickr photos.

2. Methodology. Hu et al [41] designed a three-layer framework to extract areas of interest (AOIs) from geotagged photos to understand the spatiotemporal dynamics of these areas. Tang et al [40] constructed a model of tourists' sense of place and studied their perceptions and evaluations of tourist destinations from four dimensions: natural scenery, social cultural setting, tourism function, and affectional attachment. Wang et al [5] used the kernel density estimation (KDE) algorithm to analyze tourists' attention to the landscape at multi-spatiotemporal scales. Most of the existing methods have regarded a tourist destination as an integral spatial unit for studying evolutionary rules at multi-temporal scales. While others consider multi-spatiotemporal scales, there is no correlation concerning the values between scales, which affects the accuracy of these approaches. Inspired by the existing methods, TDPMTGC fully considers the spatiotemporal scale characteristics of big data. Tourism text data granules are used to represent landscape objects in tourism geography, the multi-spatiotemporal scales in tourism GIScience are depicted by the multi-hierarchical structure of GrC, and the spatial and temporal dimensions are integrated into a systematic framework as attributes of the data granules. In this way, quantitative calculations of multi-spatiotemporal scales and popularity deduction between adjacent scales and across scales can be achieved. The potential advantages and values of this approach will be reflected by the following aspects in future applications.

① TDPMTGC has good semantic scalability. UGC data are granularized and reorganized based on spatiotemporal scales to form text data granules with clear spatiotemporal

semantics. Moreover, the granulation criteria can be extended to geography or to other thematic semantics, such as tourism emotion, sightseeing, consumption behaviors and service perceptions. Thus, this approach can not only quantitatively calculate tourist spatial popularity but can also be combined with other methods for studying tourist spatiotemporal behaviors, landscape preferences, and spatial images. TDPMTGC has a wide range of applications and can be used to support different research goals in tourism, geography or other fields of humanities and social sciences.

② TDPMTGC has good adaptability to spatiotemporal scales and types of tourist destinations. In terms of scale design, the granulation criteria of each layer are independent. The data in the upper scale are mapped to the data in lower scales through granulation criteria between each layer. Making changes in the granular layers and scale requires changing only the granulation criteria between the affected adjacent granular layers, which will not affect other granular layers. Therefore, the number of spatiotemporal scales can be adjusted dynamically based on the scale and development characteristics of tourist destinations when using TDPMTGC. For example, some tourist destinations, such as ancient cities, have no tourist routes; thus, the spatial scales could be simplified and the tourist route layer could be deleted. TDPMTGC is applicable to tourist destinations with different types and themes, for example, nature and humanity, which can facilitate comparative studies involving different types of tourist destinations.

③ TDPMTGC can be adapted to dynamic changes in the data. The granular structure of tourism text data supports the expansion of dynamic incremental data in a specific granular layer without affecting other layers. TDPMTGC can dynamically calculate TDP corresponding to the varying granular layers and achieve real-time monitoring of TDP at multi-spatiotemporal scales.

3. Experimental results. By comparing the AOI growth model, Hu et al [41] found that AOIs in developed cities have large initial areas but slow development speeds, while AOIs in rapidly developing cities have low initial values but significant growth rates. Tang et al [40] found that the natural landscape of *Jiuzhaigou* has received high perception evaluation scores and presents good general recognition by tourists, while the perception evaluation scores of its social and cultural environment are relatively low. Wang et al [5] discovered popularity routes and scenic spots in *Jiuzhaigou* by mining the spatial pattern and evolutionary processes of tourists' attention at multi-spatiotemporal scales. TDPMTGC not only obtained conclusions consistent with these previous results but also revealed detailed features of TDP that were not described in previous studies because it allows a quantitative analysis of the driving forces of tourism phenomena. These results suggest that TDPMTGC has better precision and quantitative and cross-scale calculation and deduction abilities compared with previous approaches.

## 7 Conclusions and future work

In this paper, we introduce the idea of GrC into tourism GIScience, allowing quantitative calculations of TDP to be conducted based on unstructured tourism UGC text. We propose the granular structure of multi-spatiotemporal tourism text data, design a GrC model of tourism text based on inclusion degree, and implement a text mining approach to calculate TDP based on GrC. The main contributions of TDPMTGC include the following: (1) A regularized data recombination based on granular structure is achieved for unstructured tourism UGC. This recombination includes both implicit spatial semantics and explicit temporal semantics, which can improve tourism GIScience research based on unstructured text data mining. (2) We can

describe TDP at both single spatial or temporal scale as well as the patterns and processes of TDP at multi-spatiotemporal scales using data granular layers corresponding to the spatiotemporal scales. (3) The inclusion degree based on conditional probability can be used to describe spatial popularity and standardizes basic spatiotemporal units at different spatiotemporal scales, which can be used to quantify the contribution degrees of different spatial and temporal units to TDP.

The results of the presented case study of *Jiuzhaigou* are consistent with previous results [5,40], confirming the feasibility and effectiveness of TDPMTGC. The main conclusions are as follows. (1) From the perspective of landscape preference, tourists pay more attention to the natural landscape of *Jiuzhaigou*, especially its water-related landscapes. (2) Based on the spatial characteristics of TDP, different tourist routes have different popularities: *Rizegou* and *Zechawagou* have higher popularity; *Shuzhenggou* has the lowest popularity; and *Wucaichi*, *Huohuahai* and *Nuorilangpubu* are representative landscapes with high popularity. (3) According to the temporal pattern of TDP, there are monthly differences: the peak season, with high popularity lasts from June to October, while the off-season runs from November to May of the next year. The daily variations in the popularity of attractions present three patterns in different seasons: three peaks and two valleys, two peaks and one valley, or no significant peaks and valleys. In addition, abnormal popularity peaks at yearly, monthly and daily scales were also identified in our results.

The case study of *Jiuzhaigou* reveals detailed features of TDP that have not been described in previous studies, which supports quantitative analysis of the driving forces of tourism phenomena. (1) TDP at a finer spatial scale can explain the contributions of that scale to overall popularity at the macro scale. For example, *Wucaichi* and *Changhai*, which reach the highest popularity level, make a decisive contribution to the high popularity of *Zechawagou*. This approach also accurately locates periods with abnormal popularity, such as the Dragon Boat Festival, National Day Golden week, and the *Jiuzhaigou* earthquake, from the daily tourism patterns in June and October 2013 and August 2017, providing a quantitative explanation for these abnormal phenomena. (2) A comprehensive analysis of multi-spatiotemporal scales is implemented, revealing the new tourism spatial cognition. This analysis reveals a phenomenon in which most microblog users' descriptions of *Jiuzhaigou* exist at the *Jiuzhaigou* scenic area scale, while fewer than one-quarter of users clearly describe specific tourist routes or scenic spots. The users who describe small-scale spaces typically focus on a scenic spot, which reflects a significant weakening of the tourist route scale in tourists' cognition. The comprehensive analysis at multi-temporal scales shows that the variation curves of the peak- and off-seasons are obviously different: the analysis shows a pattern of three peaks and two valleys in the peak season, while the off-season presents double peaks with a single valley or unremarkable peaks and valleys.

This paper focuses on the initial design of TDPMTGC, which constitutes an exploration of the methodology of tourism GIScience. The following issues need further research: 1) an automatic classification algorithm needs to be designed for performing data granulation; 2) the spatial position and scale features need to be accurately measured and the related parameters of TDPMTGC need to be adjusted and optimized; 3) the results of *Jiuzhaigou* need to be calculated from more perspectives, and the spatiotemporal behavior characteristics of tourists need to be analyzed at a more detailed spatiotemporal scale; and 4) more cases of different types should be tested for comparison purposes, in which the spatial patterns and evolutionary rules of tourism should be identified with respect to scenic areas, tourist destinations and larger regional scales, providing quantitative data and calculation results to support the analysis of the driving mechanisms of tourism spatial evolution.

## Supporting information

**S1 File. Supporting document for the use of dataset.**
(PDF)

## Author Contributions

**Conceptualization:** Chi Yunxian, Li Renjie.

**Data curation:** Chi Yunxian, Guo Fenghua.

**Formal analysis:** Chi Yunxian, Guo Fenghua.

**Funding acquisition:** Li Renjie.

**Investigation:** Chi Yunxian, Guo Fenghua.

**Methodology:** Chi Yunxian.

**Project administration:** Li Renjie.

**Software:** Chi Yunxian.

**Supervision:** Li Renjie, Zhao Shuliang.

**Validation:** Li Renjie, Zhao Shuliang.

**Visualization:** Chi Yunxian.

**Writing – original draft:** Chi Yunxian.

**Writing – review & editing:** Li Renjie, Zhao Shuliang.

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
