## [Decision Letter · Decision Letter 0]

17 Sep 2019

PONE-D-19-21928

Measuring multispatiotemporal scale tourism attraction based on text granular computing

PLOS ONE

Dear Dr. Renjie,

Thank you for submitting your manuscript to PLOS ONE. After careful consideration, we feel that it has merit but does not fully meet PLOS ONE’s publication criteria as it currently stands. Therefore, we invite you to submit a revised version of the manuscript that addresses the points raised during the review process.

The three expert reviewers provided constructive comments and suggestions to further improve the research. Please address them in a major revision, especially thinking about the following two major points.

(1) Contributions: e.g., what unique information can we gain by applying TAMTGC to texts, which cannot be obtained via other possibly simpler approaches.

(2) The authors did not integrate spatial and temporal scales into one systematic model; this paper is rather a multi-spatial-scale and multi-temporal-scale approach than a multi-spatiotemporal approach.

We would appreciate receiving your revised manuscript by Oct 31 2019 11:59PM. To enhance the reproducibility of your results, we recommend that if applicable you deposit your laboratory protocols in protocols.io, where a protocol can be assigned its own identifier (DOI) such that it can be cited independently in the future. For instructions see: http://journals.plos.org/plosone/s/submission-guidelines#loc-laboratory-protocols

We look forward to receiving your revised manuscript.

Kind regards,

Song Gao, Ph.D.

Academic Editor

PLOS ONE

1. In your manuscript you note, "Our research group purchased the Sina microblog commercial interface and downloaded the microblog data within the spatial range of the scenic area.." You also have provided data in the supporting information files. Please confirm in your response to reviewers that you have the necessary permissions to share the data. If the data are owned by a third party and you do not have the necessary permissions to publish the data in the Supporting Information files, in the Data Availability Statement, please (1) explain the data sharing restrictions and (2) provide sufficient information for other researchers to obtain the data in the same way you did. Also, in the Methods, ensure that you have provided sufficient details for others to be able to find the same data and replicate the analyses.

2. Please remove your figures from within your manuscript file, leaving only the individual TIFF/EPS image files, uploaded separately.  These will be automatically included in the reviewers’ PDF.

3. Please ensure that you refer to Figure 3 in your text as, if accepted, production will need this reference to link the reader to the figure.

Reviewers' comments:

Reviewer's Responses to Questions

**Comments to the Author**

1. Is the manuscript technically sound, and do the data support the conclusions?

Reviewer #1: Partly

Reviewer #2: Partly

Reviewer #3: Yes

2. Has the statistical analysis been performed appropriately and rigorously? 

Reviewer #1: N/A

Reviewer #2: Yes

Reviewer #3: Yes

3. Have the authors made all data underlying the findings in their manuscript fully available?

Reviewer #1: Yes

Reviewer #2: Yes

Reviewer #3: Yes

4. Is the manuscript presented in an intelligible fashion and written in standard English?

Reviewer #1: Yes

Reviewer #2: Yes

Reviewer #3: Yes

5. Review Comments to the Author

Reviewer #1: This paper introduces granular computing model into tourism geography to better describe tourism attractions through multispatiotemporal scales. However, I have a few questions that I hope the authors can re-consider:

The multispatiotemporal scales are not well designed or explained. The authors proposes four spatial scales: scenic spot scale, tourist route scale, scenic area scale, and tourist destination scale. However, there is no justification of why and how these four levels are chosen. Similarly, the temporal scales include year, month, day, and hour scales. The authors also did not justify the reason of choosing such temporal scales.

In Section 3.1, the spatial dimension is constructed from smaller scale to larger scale, but the temporal scale division is conducted from larger scale to smaller scale. The authors need to justify the reason.

The authors mentioned that existing approaches were treating "tourist destination as a whole unit", so this research is to provide an advanced approach by dividing the tourist destination into smaller scales. However, this research proposes a multi-spatiotemporal approach. The authors need to provide reasons why temporal scale is also included in the introduction section.

In calculating the attraction values, this research uses count of a specific scenic spot/route/etc. from the user generated text. However, the authors did not justify why 'count' is sufficient to calculate the attraction value. The authors should discuss the possibilities of other attributes as well. In addition, the uncertainties of analyzing user generated text should be discussed to justify that 'count' or other chosen attribute is positively correlated to attraction of the scenic spot/route/etc. This is because mentioning a specific scenic spot/route/etc. may not necessarily indicate their attractiveness, but may be of other reasons, e.g. accident, negative experience, hours of operation, etc.

One of my biggest concerns of this research is that the authors did not integrate spatial and temporal scales into one systematic model. As demonstrated in approach section and case study section, the approaches are dealing with the spatial scale and then the temporal scale separately. To me, this research is not proposing a multi-spatiotemporal approach, but rather a multi-spatial-scale and multi-temporal-scale approach. I would be intrigued to see spatial and temporal scales truly integrated in one granular computing model, which will make a great contribution to GIScience.

Reviewer #2: This paper proposes to use a granular computing model TAMTGC for analyzing tourism attraction based on user generated content (UGC). The authors described the methodological details of TAMTGC, and conducted an experiment based on Jiuzhaigou using Sina microblog posts.

Strong points:

- The research topic of applying granular computing to analyzing tourism attraction is interesting.

- The authors provided detailed descriptions on the methodology.

Weak point:

- The added value of TAMTGC is unclear. In other words, what unique information can we gain by applying TAMTGC to texts, which cannot be obtained via other possibly simpler approaches? For example, in lines 51-52, the authors wrote: "Tourism attraction can be expressed using the number of visitors [2-4], the index related to online search and evaluation, and the User Generated Content (UGC) published by tourists [5-7]." So what unique and additional information can we gain using TAMTGC compared with e.g., using simply the number of visitors or the number of social media posts? To address this, the authors may need to do two things. First, the authors may need to enrich the introduction section to clarify the unique information obtained by TAMTGC. Second, the authors may need to add some comparisons in their case study of Jiuzhaigou to show the additional information that can be obtained by TAMTGC.

Other more detailed issues:

- Lines 82-85: The authors may consider also discussing the following related paper on analyzing UGC for discovering interesting zones.

Hu, Y., Gao, S., Janowicz, K., Yu, B., Li, W., & Prasad, S. (2015): Extracting and understanding urban areas of interest using geotagged photos, Computers, Environment and Urban Systems, 54, 240-254.

- Line 468: "In total, we collected >100,000" It would be better to use the exact number of posts here.

- Is the dataset used in the case study all Sina microblog posts published during this period in the study area or only a sample? Please clarify.

- Table 5 has too much information and is overwhelming. Maybe the authors can highlight some values with bold font.

- Figure 4: Would the temporal variation of the attraction be similar or different from the numbers of microblog posts in the same time period? The authors may need to provide a comparison and discussion here.

- Lines 582-583: "TAMTGC can use the full volume of the tourism UGC texts, which can better reflect users' real emotional trends" ? Could the authors provide some explanation on "emotional trends" and how TAMTGC can help discover these emotional trends?

Reviewer #3: This paper presented a new method to calculate tourism attraction from textual social media data, and developed a new methodology to conduct analysis from multi-spatiotemporal scale. In general, this paper did some good contribution to spatiotemporal data mining and semantic knowledge discovery. The author claims several aspects of contributions.

The comments are as follows:

1. It is better to add a new section “Literature Review” or “Existing Work” to summary previous research on related method of semantic knowledge discovery in GIScience, related spatiotemporal data mining method, tourism attraction analysis, granular computing model, etc. And then reorganize the section of Introduction.

2. The paper claims 5 aspects of contributions in introduction section. In my opinion, some contributions are not significant enough. For example, the 4th item “TAMTGC is extensible” cannot be thought of as a contribution. And Item 1 and 2 can be combined to illustrate the contribution of TAMTGC. Item 5 should be modified to claim the TAMTGC model was successfully applied in Jiuzhaigou area to obtain some new insightful research conclusion of tourist attractions in this area.

3. In Section 3, some formulas are very long and not very readable. Especially, in some sentences, some formulas have to be inserted, which makes readers confusing. For example, “the total number of A, B and C of 49 scenic spots”, and “the attraction of A of Zhenzhutanpubu is XXXXX”. A suggestion is, the authors can replace some formulas with simple symbols (use letters A, B, C, or use simple words), and use these simple symbols in sentences when complex formulas have to appear.

4. In Section 4.2, the result of spatial scale is described using table including different place names as rows. It would be better to use maps to obtain better result visualization effects. Especially, most readers are not familiar with where Jiuzhaigou is, and where the locations of different tourist spots are. So a map of Jiuzhaigou describing locations of different travel spots could be helpful.

5. From Table 2-5, it could be found that most calculation results are VERY small between 0.0000 and 0.0100. Can the authors consider some data normalization method, to normalize the intermediate data and final results to a value between 0.0 and 1.0, or a tourist attraction score between 0.0 and 100.0?

6. Some word and grammar errors can be found. There is a logic error in the FIRST sentence of this paper. It should be “tourism GIScience mainly studies a series of basic problems in XXXXX …” During my review of this paper, more than 10 grammar errors were found, including tense inconsistency and preposition errors. In addition, “multi-spatiotemporal” should be used instead of “multispatiotemporal”. When using “multi” with other nouns, there should always be a “-“ between them.

6. PLOS authors have the option to publish the peer review history of their article (what does this mean?). If published, this will include your full peer review and any attached files.

Reviewer #1: No

Reviewer #2: No

Reviewer #3: No

---

## [Author Response · Author response to Decision Letter 0]

20 Nov 2019

Responses to reviewers concerning the revision of manuscript [PONE-D-19-21928] (Measuring multi-spatiotemporal scale tourist destination popularity based on text granular computing)

Dear editors and reviewers,

Thank you very much for your comments and suggestions on our manuscript to further improve the presentation of our research. After carefully reading, thinking on and re-considering the comments, we found that your questions were of great help for improving this manuscript and the suggestions are good and quite pertinent. Regarding the opinions concerning the revisions, we have made substantial modifications to this manuscript that we believe are advantageous to our research. We thank you again for your comments and suggestions. In addition, I would like to thank all the review experts and editors for their recognition of our manuscript. Our specific modifications and explanations concerning the manuscript are listed below for each reviewer and each comment:

Academic editor:

Comment 1:

At this time, please confirm in your response to reviewers that you have the necessary permissions to share the data. If the data are owned by a third party and you do not have the necessary permissions to publish the data in the Supporting Information files: please (1) explain the data sharing restrictions and (2) please clarify whether or not you had special access privileges to the data that others did not have. (3) please also provide us with information for a non author point of contact that interest researchers may contact regarding data access.

Revision:

(1) The data sharing restrictions.

We have signed an agreement with a third party. In view of the particularity of commercial data, the data in this paper are only used for scientific research exchange.

(2) Please clarify whether or not you had special access privileges to the data that others did not have.

We have special access privileges to the data that others did not have. Because the commercial data are not public, the data used in this paper were downloaded by purchasing the Sina microblog commercial interface, the full amount of microblog data about the research area could be obtained during the purchase period, while only a small amount of public data could be downloaded without purchase.

(3) Please also provide us with information for a non author point of contact that interest researchers may contact regarding data access.

The data we purchased have been downloaded and saved on our hard drive and are accessible at any time, and we have submitted the dataset to Journal of PLOS ONE. We have asked the data administrator to agree that the data in this paper can be shared on the premise of scientific research exchange. Journal of PLOS ONE has access to our minimal dataset and can serve as a non author point of contact for these queries. Therefore, interested researchers can contact our corresponding authors or Journal of PLOS ONE for access to the data. Researchers who obtain data are asked not to spread the data widely.

The data administrator also has access to our minimal dataset and can serve as a non author point of contact for these queries. Moreover, if researchers would like additional data, they can contact the data administrator to make a purchase:

Data administrator: Youzhi Liu

Telephone: 008610-60619366

Email: youzhi@staff.weibo.com

Please note that due to the dynamic nature of microblog data, microblog users may delete some data; therefore, the data downloaded in different periods may vary but will not differ overall.

Comment 2:

Please do not include funding sources in the Acknowledgments or anywhere else in the manuscript file. Funding information should only be entered in the financial disclosure section of the submission system.

Revision:

We apologize for adding extra information in the manuscript. We have deleted the funding sources in the Acknowledgments.

Thank you very much for your patient guidance and help. If there are any mistakes that need to be corrected, please let us know and we will correct them carefully and timely.

Comment 3:

Before we can proceed with your paper, please address the following queries:

a) Please confirm that the data you submitted is your 'minimal data set', which PLOS defines as consisting of the data set used to reach the conclusions drawn in the manuscript with related metadata and methods, and any additional data required to replicate the reported study findings in their entirety. This includes:

3) The points extracted from images for analysis.

b) Please confirm that you have permission to publish these data under a CC BY 4.0 license.

Revision:

a) The data we submitted is our 'minimal data set', which PLOS defines as consisting of the data set used to reach the conclusions drawn in the manuscript with related metadata and methods, and any additional data required to replicate the reported study findings in their entirety. This includes:

3) The points extracted from images for analysis.

b) We have permission to publish these data under a CC BY 4.0 license.

Reviewer #1:

Comment 1:

The multi-spatiotemporal scales are not well designed or explained. The authors proposes four spatial scales: scenic spot scale, tourist route scale, scenic area scale, and tourist destination scale. However, there is no justification of why and how these four levels are chosen. Similarly, the temporal scales include year, month, day, and hour scales. The authors also did not justify the reason of choosing such temporal scales.

Revision:

Thank you for your constructive comments. Multi-spatiotemporal scales that are not well designed or explained would directly affect the reader's understanding of the framework structure of this paper. Your comments have played an important role in improving our manuscript. Thank you for your constructive suggestions.

To better explain the multi-spatiotemporal scales of the TDPMTGC method, we have revised the contents of Section 3.2.2 “Granular structure of tourism text data” and extended the space and time scales to multiple scales rather than a fixed four-level scale. The multi-spatiotemporal scale granular structure of tourism text data is represented by the complete graph shown in Fig 1(a), in which layers of the multi-spatial granular structure correspond to the scales. The data granules in the upper scale are transformed into those in the lower scale using the granulation criteria . The data granules decrease as the scale decreases. Similarly, layers of the multi-temporal granular structure correspond to the scales, and granules in the upper scale are transformed into those in the lower scale using the granulation criteria . A complete graph represents the existence of an edge (i.e., a correlation) between any spatial-spatial, temporal-temporal, or spatial-temporal scales. There are edges among the spatial-spatial scales, edges among the temporal-temporal scales, and edges among the spatial-temporal scales; thus, the total number of edges is . The correlation between temporal scales is presupposed by the "spatial-temporal" correlation (i.e., the correlation between two temporal scales ‘ — ’ for a spatial scale is obtained by granulating in layers and , which yields the correlations ‘ — ’ and ‘ — ’). The granular structure of tourism text data can be used not only to mine features of small-scale landscapes (where represents a tourist destination) over a short period (such as when represents an annual scale) but also to mine the life cycle evolutionary laws at large scales (where represents a national or even a global scale) over long periods (such as when represents several centuries (if the data are available)).

Common spatial scales are implemented in tourist GIScience, such as scenic spots, tourist routes, scenic areas, tourist destinations, provinces, nations, etc. Similarly, common temporal scales are implemented, such as year, month, week, day, hour, minute, and second. The number of spatial and temporal scales should be selected according to the size of the tourist destination (i.e., smaller scenic areas can skip the tourist route scale). In this paper, we use four scales in the spatial dimension, namely, "scenic spot—tourist route—scenic area—tourist destination", and four scales in the temporal dimension, namely, "year—month—day—time", as examples to introduce the dataset construction method of spatial and temporal dimension.

Consequently, in Section 3 “Theory and method”, the spatial scales are no longer limited to four levels (scenic spot scale, tourist route scale, scenic area scale, and tourist destination scale) and the temporal scales are no longer limited to four levels (year, month, day and hour). However, in Section 4 (the tourist destination popularity computing approach based on granular computing model section) and Section 5 (the experimental section—a case study from Jiuzhaigou), we selected four scales each for the spatial and temporal dimensions as an example to clearly describe the approach for constructing the granular computing model dataset and the results from applying the TDPMTGC model to Jiuzhaigou because these are the usual spatial and temporal scales selected in this field.

Please refer to lines 283–304 on pages 12–13 and lines 397–406 on page 17 of the revised ‘Manuscript’ and ‘Revised Manuscript with Track Changes’.

Comment 2:

In Section 3.1, the spatial dimension is constructed from smaller scale to larger scale, but the temporal scale division is conducted from larger scale to smaller scale. The authors need to justify the reason.

Revision:

Thank you very much for your constructive comments. If the construction approach of the granular computing model dataset is not clearly explained, it could confuse readers. Thus, your comments played an important role in improving the manuscript. Thank you for your constructive suggestions.

The spatial information (such as toponymy) in multi-scale unstructured UGC data is implicit in the text and needs to be identified layer by layer. Moreover, the data granules in the lower layer are subsets of those in the next highest layer and a number of cross-scale layers (i.e., tourist route granules at a tourist route scale not only include single spot granules but also single route with multiple spots granules and multiple routes with multiple spots granules at the scenic spot scale. Similarly, they include single-route and multiple-route granules at a tourist route scale). After completing the construction of granules in the lower layer, they can be directly integrated into the granules in the upper layer, thus expanding to larger granules layer by layer. Because of this inclusion relationship between scales in the spatial dimension, the dataset is constructed from bottom to top using a scale from small to large and a granular scale that moves from fine to coarse. The temporal information in UGC data is explicit in each text; thus, data granules in lower layers inherit the labels of those in the upper layers (for example, a granule at a monthly scale must belong to a certain granule at a yearly scale). We adopt a tree structure to complete the construction of the data granules in the upper layer and then decompose them downward layer by layer. This approach clearly indicates the inheritance relationship among the data granules of each layer. Hence, in the temporal dimension, based on the spatial dataset, the dataset is constructed from top to bottom using a scale from large to small and a granular scale that moves from coarse to fine.

Please refer to lines 380–396 on pages 16–17 of the revised ‘Manuscript’ and ‘Revised Manuscript with Track Changes’.

Comment 3:

The authors mentioned that existing approaches were treating "tourist destination as a whole unit", so this research is to provide an advanced approach by dividing the tourist destination into smaller scales. However, this research proposes a multi-spatiotemporal approach. The authors need to provide reasons why temporal scale is also included in the introduction section.

Revision:

Thank you very much for your constructive comment.

Research in the field of tourism geography usually includes a time scale. Although the previous popularity analysis methods of tourist destination made multi-scale divisions on the temporal scale, they regarded a tourist destination as an integral unit on the spatial scale and often ignored its internal spatial characteristics, which affected the precision of the method. In-depth analysis of the spatiotemporal characteristics between scales helps improve model precision. However, establishing an accurate relationship between text and spatial units of different scales and integrating multi-spatial and multi-temporal scales into a systematic model are still obstacles in the study of tourism GIScience.

To accurately granulate the spatial and temporal information of tourism text, a tourism text data granule is used to represent a landscape object, which is a unified whole that possesses multiple attributes, such as spatial and temporal dimensions. The multi-spatiotemporal scales are characterized by the multi-hierarchical structure of GrC, and the transformations of granular layers and data granule size are realized by the scale selection in spatial and temporal dimensions. Therefore, all scales between the spatial and temporal dimension are related, thus making the data granules of all spatial-spatial, temporal-temporal and spatial-temporal layers comparable. This approach achieves a quantitative description and comparison of the popularity value of granules between adjacent scales and cross-scales. Therefore, the tourist destination popularity with multi-spatiotemporal scales can be calculated in a systematic framework.

Your comments played an important role in improving the manuscript. Thank you for your constructive suggestions.

Please refer to lines 81–88 and 101–112 on pages 4–5 of the revised ‘Manuscript’ and ‘Revised Manuscript with Track Changes’.

Comment 4:

In calculating the attraction values, this research uses count of a specific scenic spot/route/etc. from the user generated text. However, the authors did not justify why 'count' is sufficient to calculate the attraction value. The authors should discuss the possibilities of other attributes as well. In addition, the uncertainties of analyzing user generated text should be discussed to justify that 'count' or other chosen attribute is positively correlated to attraction of the scenic spot/route/etc. This is because mentioning a specific scenic spot/route/etc. may not necessarily indicate their attractiveness, but may be of other reasons, e.g. accident, negative experience, hours of operation, etc.

Revision:

Thank you very much for your constructive comment.

We are sorry that your understanding of this paper differs from the meaning we want to express due to our unclear explanation. We want to convey the concept of measuring multi-spatiotemporal scale ‘tourist destination popularity’ based on text granular computing, namely, tourists' attention to the landscape at different spatiotemporal scales. The number of texts published by tourists about a tourist destination reflects their attention to the landscape of that destination at different spatiotemporal scales. Therefore, we can use the total mentions of a specific scenic spot/route/etc. from the user-generated text to calculate ‘popularity’ values. When we wrote the paper, we mistakenly used the word ‘attraction’ to mean ‘popularity’, which may have caused the ambiguity concerning this topic. We have modified the whole paper and changed ‘tourism attraction’ to ‘tourist destination popularity (TDP)’.

The text of posts published by tourists are ‘counted’ to reflect the ‘tourist destination popularity’ at various spatiotemporal scales. This ‘popularity’ includes both positive and negative impressions (e.g., accidents, negative experiences, insufficient hours of operation, etc.). For example, in Section 4.3, several abnormal popularity months were found through the monthly scale popularity variation tendency model, and the driving factors of the abnormal popularity months were then further analyzed through the daily scale popularity distribution model. The abnormal popularity in June 2013 occurred mainly from the 10th to 13th, which overlapped with the Dragon Boat Festival holiday, i.e., the second holiday in which the free expressway was implemented in October 2012, thus intensifying tourists' desire to travel and leading to a sudden increase in tourist destination popularity. The abnormal popularity in October mainly lasted from the 2nd to 6th, coinciding with the National Day holiday. The popularity reached its highest value on October 2, which corresponded to the large-scale tourist detention event on that day. The daily variation in the August anomaly in 2017 with the highest peak (from August 8-11) in Fig 4(b2) coincided with the period of the 7.0-magnitude earthquake in Jiuzhaigou County on August 8, indicating that the disaster event was the main factor leading to the increase in popularity in Jiuzhaigou during this period.

Therefore, the popularity calculation method proposed in this paper can be used to identify periods of unusual attention from tourists that are coupled with holidays, special policies, tourism events and sudden disasters, thus providing a quantitative explanation for these abnormal phenomena.

Your comments played an important role in improving the manuscript. Thank you for your constructive suggestions.

Please refer to the full revised ‘Manuscript’ and ‘Revised Manuscript with Track Changes’.

Comment 5:

One of my biggest concerns of this research is that the authors did not integrate spatial and temporal scales into one systematic model. As demonstrated in approach section and case study section, the approaches are dealing with the spatial scale and then the temporal scale separately. To me, this research is not proposing a multi-spatiotemporal approach, but rather a multi-spatial-scale and multi-temporal-scale approach. I would be intrigued to see spatial and temporal scales truly integrated in one granular computing model, which will make a great contribution to GIScience.

Revision:

Thank you very much for your constructive comments. Your concerns regarding this research about integrating spatial and temporal scales into one systematic model is the core of TDPMTGC. Your comments played an important role in improving the model. Thank you for your constructive suggestions.

In response, we provide a brief introduction in the Abstract and Instruction. We also modified Section 3.2.2 to explain how the granular computing model integrates spatial and temporal scales. Finally, in the case study section (Section 5.2 ‘Tourist destination popularity mining at multi-spatiotemporal scales’), we added an experimental verification of the system model.

(1) Abstract and Instruction

To accurately granulate the spatial and temporal information of tourism text, tourism text data granules are used to represent landscape objects. These granules are unified objects that possess multiple attributes, such as spatial and temporal dimensions. The multi-spatiotemporal scales are characterized by the multi-hierarchical structure of granular computing, and transformations of granular layers and data granule size are achieved by scale selection in the spatial and temporal dimensions. Therefore, all scales between the spatial and temporal dimension are related, which allows for the comparability of the data granules of all spatial-spatial, temporal-temporal and spatial-temporal layers. This approach achieves a quantitative description and comparison of the popularity value of granules between adjacent scales and cross-scales. Therefore, the TDP with multi-spatiotemporal scales can be deduced and calculated in a systematic framework.

(2) Section 3.2.2.

First, by defining the structure of a "tourism text data granule", we show how the spatial and temporal dimensions are integrated into a single systematic model as attributes of the data granules. A tourism text data granule is a complete entity with multiple attributes, such as space and time, which must be described from spatial, temporal and other dimensions. Among them, both the spatial and temporal dimensions contain multiple scales; thus, the multi-scale structure of granules corresponds to these multi-spatiotemporal scales (see Fig 1(a)). Using this approach, the time and space dimensions are integrated into a single systematic model reflected as attributes of data granules. To describe the spatiotemporal characteristics of the data granules, it is necessary to clearly indicate their spatiotemporal scale, which can be divided into the following situations: ① To describe the characteristics of data granules at a particular spatiotemporal scale, it is necessary to fix the spatial and temporal scales of the granules (see Fig 1(b)); ② To describe the characteristics of data granules at a specific spatial (or temporal) scale, it is necessary to fix the spatial (or temporal) scale of the granules and mine the evolution rules of granules at that multi-temporal (or multi-spatial) scale (see Fig 1(c) and 1(d)); and ③ To describe the characteristics of data granules at multi-spatiotemporal scales, multiple scales of the spatial and temporal dimensions of the granules should be selected to perform comprehensive mining (see Fig 1(e)).

Then, we describe the implementation of the multi-spatiotemporal scale granular structure. The multi-spatiotemporal scale granular structure of tourism text data is represented by the complete graph shown in Fig 1(a), in which layers of the multi-spatial granular structure correspond to the scales. The data granules in the upper scale are transformed into those in the lower scale using the granulation criteria . The data granules decrease as the scale decreases. Similarly, layers of the multi-temporal granular structure correspond to the scales, and granules in the upper scale are transformed into those in the lower scale using the granulation criteria . A complete graph represents the existence of an edge (i.e., a correlation) between any spatial-spatial, temporal-temporal, or spatial-temporal scales. There are edges among the spatial-spatial scales, edges among the temporal-temporal scales, and edges among the spatial-temporal scales; thus, the total number of edges is . The correlation between temporal scales is presupposed by the "spatial-temporal" correlation (i.e., the correlation between two temporal scales ‘ — ’ for a spatial scale is obtained by granulating in layers and , which yields the correlations ‘ — ’ and ‘ — ’). The granular structure of tourism text data can be used not only to mine features of small-scale landscapes (where represents a tourist destination) over a short period (such as when represents an annual scale) but also to mine the life cycle evolutionary laws at large scales (where represents a national or even a global scale) over long periods (such as when represents several centuries (if the data are available)). According to the actual needs, subgraphs can be extracted from Fig 1(a) to achieve landscape law mining at a single-space/single-time scale (see Fig 1(b)), single-space/multiple-time scales (see Fig 1(c)), multiple-space/single-time scales (see Fig 1(d)), and multiple-space/multiple-time scales (see Fig 1(e)).

In conclusion, a tourism text data granule is a unified whole possessing multiple attributes, such as a spatial and a temporal dimension. The transformations of granular layers and data granule size are achieved by scale selection in both the spatial and temporal dimensions. Therefore, all the scales between spatial and temporal dimension are related, which allows for the comparability of the data granules of all spatial-spatial, temporal-temporal and spatial-temporal layers. This approach allows for comparisons of the popularity value of data granules both among adjacent scales and across scales, forming unique information that we can gain by applying TDPMTGC to texts that cannot be obtained via other, possibly simpler, approaches (e.g., simply counting the number of visitors or the number of social media posts). We can analyze the geographic spatiotemporal relations among the multiple granular layers using the granular structure . Thus, is a useful tool for finely describing the multi-spatiotemporal patterns of tourist destination popularity.

(3) In the case study part.

Fig 3 is taken as an example to demonstrate the systematic model from three aspects, namely, ‘The spatiotemporal model associated with scenic areas’, ‘The spatiotemporal model associated with tourist routes’, and ‘The spatiotemporal model associated with scenic spots’. Finally, we conclude that TDPMTGC makes the data granules of all spatial-spatial, temporal-temporal and spatial-temporal layers comparable and then achieves the comparison of popularity values of data granules between adjacent scales and across scales. Detailed and quantitative descriptions of tourist destination popularity at multi-spatiotemporal scales are helpful for comprehensively and deeply exploring the spatiotemporal characteristics of tourism from the viewpoint of tourists' cognition.

Please refer to lines 29–38 on page 2, lines 101–112 on page 5, lines 253–258 and 267–315 on pages 11–12, and lines 595–798 on pages 26–36 of the revised ‘Manuscript’ and ‘Revised Manuscript with Track Changes’.

Reviewer #2:

Comment 1:

Weak point:

- The added value of TAMTGC is unclear. In other words, what unique information can we gain by applying TAMTGC to texts, which cannot be obtained via other possibly simpler approaches? For example, in lines 51-52, the authors wrote: "Tourism attraction can be expressed using the number of visitors [2-4], the index related to online search and evaluation, and the User Generated Content (UGC) published by tourists [5-7]." So what unique and additional information can we gain using TAMTGC compared with e.g., using simply the number of visitors or the number of social media posts? To address this, the authors may need to do two things. First, the authors may need to enrich the introduction section to clarify the unique information obtained by TAMTGC. Second, the authors may need to add some comparisons in their case study of Jiuzhaigou to show the additional information that can be obtained by TAMTGC.

Revision:

Thank you very much for your constructive comment.

The values added by TDPMTGC are described in 4 parts of the paper.

(1) Abstract

To accurately granulate the spatial and temporal information of tourism text, tourism text data granules are used to represent landscape objects. These granules are unified objects that possess multiple attributes, such as spatial and temporal dimensions. The multi-spatiotemporal scales are characterized by the multi-hierarchical structure of granular computing, and transformations of granular layers and data granule size are achieved by scale selection in the spatial and temporal dimensions. Therefore, all scales between the spatial and temporal dimension are related, making the data granules of all spatial-spatial, temporal-temporal and spatial-temporal layers comparable. This approach achieves a quantitative description and comparison of the popularity value of granules between adjacent scales and cross-scales. Therefore, the TDP with multi-spatiotemporal scales can be deduced and calculated in a systematic framework.

(2) Introduction and Section 3.2.2

"... a tourism text data granule is used to represent a landscape object, which is a unified whole that possesses multiple attributes, such as spatial and temporal dimensions. The multi-spatiotemporal scales are characterized by the multi-hierarchical structure of GrC, and the transformations of granular layers and data granule size are realized by the scale selection in spatial and temporal dimensions. Therefore, all scales between the spatial and temporal dimension are related, which allows for the comparability of the data granules of all spatial-spatial, temporal-temporal and spatial-temporal layers. This approach achieves a quantitative description and comparison of the popularity value of granules between adjacent scales and cross-scales. Therefore, the tourist destination popularity with multi-spatiotemporal scales can be calculated in a systematic framework. Thus, we can gain unique information by applying TDPMTGC to texts that cannot be obtained via other, possibly simpler, approaches (e.g., simply counting the number of visitors or the number of social media posts)".

(3) A case study from Jiuzhaigou:

In Section 5.2 of this paper, we describe in detail how the TDPMTGC method can achieve quantitative comparisons of the popularity values of data granules between adjacent scales and across scales, meaning that the unique information obtained by TDPMTGC can be compared with that of other methods.

Finally, we could draw a conclusion that TDPMTGC makes the data granules of all spatial-spatial, temporal-temporal and spatial-temporal layers comparable and then achieves the comparison of popularity values of data granules between adjacent scales and across scales. Detailed and quantitative descriptions of tourist destination popularity at multi-spatiotemporal scales are helpful for comprehensively and deeply exploring the spatiotemporal characteristics of tourism from the viewpoint of tourists' cognition

(4) Discussion

In the Discussion, we compare TDPMTGC with 3 approaches in the paper, including the one mentioned by the reviewer, and the advantages of our method and the unique information it offers are illustrated.

First, this paper compares three relevant research approaches to illustrate the inheritance and further innovation of TDPMTGC based on existing approaches and how it will lead to further research.

TDPMTGC has good adaptability to spatiotemporal scales and types of tourist destinations. In terms of scale design, the granulation criteria of each layer are independent. The data in the upper scale are mapped to the data in lower scales through granulation criteria between each layer. Making changes in the granular layers and scale requires changing only the granulation criteria between the affected adjacent granular layers, which will not affect other granular layers. Therefore, the number of spatiotemporal scales can be adjusted dynamically based on the scale and development characteristics of tourist destinations when using TDPMTGC. For example, some tourist destinations, such as ancient cities, have no tourist routes; thus the spatial scales could be simplified and the tourist route layer could be deleted. TDPMTGC is applicable to tourist destinations with different types and themes, for example, nature and humanity, which can facilitate comparative studies involving different types of tourist destinations. 

TDPMTGC can be adapted to dynamic changes in the data. The granular structure of tourism text data supports the expansion of dynamic incremental data in a specific granular layer without affecting other layers. TDPMTGC can dynamically calculate tourist destination popularity corresponding to the varying granular layers and achieve real-time monitoring of tourist destination popularity at multi-spatiotemporal scales.

Your comments played an important role in improving the manuscript. Thank you for your constructive suggestions.

Please refer to lines 29–38 on page 2, lines 101–112 on page 5, lines 305–315 on page 13, lines 595–798 on pages 26–36 and lines 850–864 on page 38 of the revised ‘Manuscript’ and ‘Revised Manuscript with Track Changes’.

Comment 2:

- Lines 82-85: The authors may consider also discussing the following related paper on analyzing UGC for discovering interesting zones.

Hu, Y., Gao, S., Janowicz, K., Yu, B., Li, W., & Prasad, S. (2015): Extracting and understanding urban areas of interest using geotagged photos, Computers, Environment and Urban Systems, 54, 240-254.

Revision:

Thank you very much for your constructive comment.

In the discussion section, we compare TDPMTGC with three related research approaches, including one mentioned above.

Previous tourist destination popularity research approaches have important implications for this paper. We take the approaches of Hu [41], Wang [5], Tang [40] as examples and compare them with the TDPMTGC proposed in this paper to both acknowledge the inheritance TDPMTGC owes to the existing approaches as well as its further innovations and to reflect its potential advantages and value in future applications, leading to further research questions.

(1) Dataset. Before the advent of the big data era, questionnaires represented the main method of obtaining user data (e.g., Tang et al [40]). However, the rapid development of the Internet has caused the data scale to explode. Increasingly, scholars focus on mining social media data, such as Flickr photos and microblog data (e.g., Hu et al [41] and Wang et al [5]). TDPMTGC uses the full content of tourism UGC texts, which contain rich spatiotemporal and semantic information that is conducive to in-depth explorations of the rules governing tourists' spatiotemporal behaviors and analysis of the driving mechanisms of tourism spatial patterns and processes. This approach better reflects users' real emotional trends than does data collected based on specific research objectives, such as questionnaire surveys and interviews, and it reduces the differences caused by sparse or inconsistent samples. For example, analyzing the variation tendency of popularity of Jiuzhaigou at the daily scale, we find that the unusual period of attention by tourists is associated with holidays, special policies, tourism events and sudden disasters. The feature extraction of tourism UGC text from an abnormal time period can be used to analyze users' emotional trends. One advantage of TDPMTGC is that the data types it can use are unrestricted. Although we chose text for this study, other types of data could also be employed, and we plan to conduct further research using Flickr photos.

(2) Methodology. Hu et al [41] designed a three-layer framework to extract areas of interest (AOIs) from geotagged photos to understand the spatiotemporal dynamics of these areas. Tang et al [40] constructed a model of tourists' sense of place and studied their perceptions and evaluations of tourist destinations from four dimensions: natural scenery, social cultural setting, tourism function, and affectional attachment. Wang et al [5] used the kernel density estimation (KDE) algorithm to analyze tourists' attention to the landscape at multi-spatiotemporal scales. Most of the existing methods have regarded a tourist destination as an integral spatial unit for studying evolutionary rules at multi-temporal scales. While others consider multi-spatiotemporal scales, there is no correlation concerning the values between scales, which affects the accuracy of these approaches. Inspired by the existing methods, TDPMTGC fully considers the spatiotemporal scale characteristics of big data. Tourism text data granules are used to represent landscape objects in tourism geography, the multi-spatiotemporal scales in tourism GIScience are depicted by the multi-hierarchical structure of GrC, and the spatial and temporal dimensions are integrated into a systematic framework as attributes of the data granules. In this way, quantitative calculations of multi-spatiotemporal scales and popularity deduction between adjacent scales and across scales can be achieved. The potential advantages and values of this approach will be reflected by the following aspects in future applications.

① TDPMTGC has good semantic scalability. UGC data are granularized and reorganized based on spatiotemporal scales to form text data granules with clear spatiotemporal semantics. Moreover, the granulation criteria can be extended to geography or to other thematic semantics, such as tourism emotion, sightseeing, consumption behaviors and service perceptions. Thus, this approach can not only quantitatively calculate tourist spatial popularity but can also be combined with other methods for studying tourist spatiotemporal behaviors, landscape preferences, and spatial images. TDPMTGC has a wide range of applications and can be used to support different research goals in tourism, geography or other fields of humanities and social sciences.

② TDPMTGC has good adaptability to spatiotemporal scales and types of tourist destinations. In terms of scale design, the granulation criteria of each layer are independent. The data in the upper scale are mapped to the data in lower scales through granulation criteria between each layer. Making changes in the granular layers and scale requires changing only the granulation criteria between the affected adjacent granular layers, which will not affect other granular layers. Therefore, the number of spatiotemporal scales can be adjusted dynamically based on the scale and development characteristics of tourist destinations when using TDPMTGC. For example, some tourist destinations, such as ancient cities, have no tourist routes; thus, the spatial scales could be simplified, and the tourist route layer could be deleted. TDPMTGC is applicable to tourist destinations with different types and themes, for example, nature and humanity, which can facilitate comparative studies involving different types of tourist destinations.

③ TDPMTGC can be adapted to dynamic changes in the data. The granular structure of tourism text data supports the expansion of dynamic incremental data in a specific granular layer without affecting other layers. TDPMTGC can dynamically calculate tourist destination popularity corresponding to the varying granular layers and achieve real-time monitoring of tourist destination popularity at multi-spatiotemporal scales.

(3) Experimental results. By comparing the AOI growth model, Hu et al [41] found that AOIs in developed cities have large initial areas but slow development speeds, while AOIs in rapidly developing cities have low initial values but significant growth rates. Tang et al [40] found that the natural landscape of Jiuzhaigou has received high perception evaluation scores and presents good general recognition by tourists, while the perception evaluation scores of its social and cultural environment are relatively low. Wang et al [5] discovered popularity routes and scenic spots in Jiuzhaigou by mining the spatial pattern and evolutionary processes of tourists' attention at multi-spatiotemporal scales. TDPMTGC not only obtained conclusions consistent with these previous results but also revealed detailed features of tourist destination popularity that were not described in previous studies because it allows a quantitative analysis of the driving forces of tourism phenomena. These results suggest that TDPMTGC has better precision and quantitative and cross-scale calculation and deduction abilities compared with previous approaches.

Your comments played an important role in improving the manuscript. Thank you for your constructive suggestions.

Please refer to lines 804–876 on pages 36–39 of the revised ‘Manuscript’ and ‘Revised Manuscript with Track Changes’.

Comment 3:

- Line 468: "In total, we collected >100,000" It would be better to use the exact number of posts here.

Revision:

Thank you very much for your constructive comment.

The precision of the paper was improved by using more accurate numbers. We revised ‘>100,000’ to the exact number of posts, i.e., 105,226.

Your comments played an important role in improving the manuscript. Thank you for your constructive suggestions.

Please refer to line 590 on page 26 of the revised ‘Manuscript’ and ‘Revised Manuscript with Track Changes’.

Comment 4:

- Is the dataset used in the case study all Sina microblog posts published during this period in the study area or only a sample? Please clarify.

Revision:

Thank you very much for your constructive comment.

We apologize for being unclear in our description of the dataset in the original paper and possibly confusing readers. We have clarified the dataset in the revised manuscript as follows: In total, we collected 105,226 microblog posts from 2013 to 2017, which constitutes all the Sina microblog posts published during this period regarding Jiuzhaigou. By filtering noise data (the number of noise data is 68,486), we obtained 36,740 valid tourism text entries (see Table 1) that constitute the dataset.

Your comments played an important role in improving the manuscript. Thank you for your constructive suggestions.

Please refer to lines 589–593 on page 26 of the revised ‘Manuscript’ and ‘Revised Manuscript with Track Changes’.

Comment 5:

- Table 5 has too much information and is overwhelming. Maybe the authors can highlight some values with bold font.

Revision:

Thank you very much for your constructive comment.

We agree that tables with too much information are overwhelming without bold font and will confuse readers. In this revision, we used bold font and underlined text to highlight the three levels of popularity spots. Three scenic spots in the first level have the highest popularity: Wucaichi and Changhai in Zechawagou and Wuhuahai in Rizegou (in bold underlined font). The scenic spots in the second level with high popularity are concentrated in Rizegou, including the 7 scenic spots of Zhenzhutanpubu, Jianzhuhai, Nuorilangpubu, Xiongmaohai, Yuanshisenlin, Jinghai and Zhenzhutan (in bold font). The scenic spots in Shuzhenggou are ranked only at the third level and include Luweihai, Huohuahai, Shuzhengzhai, Laohuhai and Xiniuhai (underlined font).

Your comments played an important role in improving the manuscript. Thank you for your constructive suggestions.

Please refer to line 611 on pages 27–28 and lines 701–707 on page 32 of the revised ‘Manuscript’ and ‘Revised Manuscript with Track Changes’.

Comment 6:

- Figure 4: Would the temporal variation of the attraction be similar or different from the numbers of microblog posts in the same time period? The authors may need to provide a comparison and discussion here.

Revision:

Thank you very much for your constructive comment.

To explain this problem clearly, we added Section 5.2.4 "The relationship between popularity variation tendency and the numbers of microblog posts" for comparison and discussion.

The temporal variation of the TDP is calculated based on the numbers of microblog posts during the same time period. The two variations are similar but not identical, and there are three main differences.

(1) Source data and reorganized data. The temporal variations in TDP as calculated by TDPMTGC are based on the text dataset after data reorganization rather than on the source data of microblog posts during the same period of the research area. Taking the data in 2017 as an example, 20,764 pieces of source data were focused on Jiuzhaigou in 2017, although this number was reduced to 7,277 after data reorganization. A comparison of the daily variation patterns within months (see Fig 4(d1)) showed that their overall trend was consistent and both were affected by the earthquake in Jiuzhaigou on August 8. However, the source data contain texts that are unrelated to the research area; thus, the variations are not exactly the same.

(2) Intersections between data granules. Intersections occur between data granules at some spatial scales, and the intersecting parts of the text belong to multiple granules. When calculating the comprehensive popularity of data granules, it is necessary to include the intersecting parts of the text in multiple granules at the same time, resulting in a text expansion compared with the source data, and these variations are slightly different from the changing trends in the number of microblog posts. For example, the route granules at the tourist route scale include a single spot, one route with multiple spots, multiple routes with multiple spots, single route and multiple routes, among which multiple routes with multiple spots and multiple routes granules simultaneously belong to multiple route granules. Therefore, the absolute number of routes is slightly different from the overall number of microblog posts (see Fig 4(d2)).

(3) The popularity value of the same data granules can be different at different scales. For example, the popularity of Wucaichi at the scenic spot scale is 18.91% as calculated based on scenic spots, while its comprehensive popularity in the scenic area is 5.85% (see Table 5). Moreover, due to the different inclusion relationships of data granules at different scales, there is not necessarily a proportional relationship between the popularity values (i.e., multiple routes with multiple spots granules belong to multiple tourist route granules at the tourist route scale but only to one granule in the scenic area scale and thus are calculated differently on different scales).

Your comments played an important role in improving the manuscript. Thank you for your constructive suggestions.

Please refer to lines 763–792 on pages 34–35 of the revised ‘Manuscript’ and ‘Revised Manuscript with Track Changes’.

Comment 7:

- Lines 582-583: "TAMTGC can use the full volume of the tourism UGC texts, which can better reflect users' real emotional trends"? Could the authors provide some explanation on "emotional trends" and how TAMTGC can help discover these emotional trends?

Revision:

Thank you very much for your constructive comment.

We only briefly mentioned in the discussion section of the article that "TDPMTGC can use the full volume of the tourism UGC texts, which can better reflect users' real emotional trends" and did not provide a further explanation, which was an oversight. Therefore, we added a more complete explanation of this potential advantage. ‘For example, analyzing the variation tendency of popularity of Jiuzhaigou at the daily scale, we find that the unusual period of attention by tourists is associated with holidays, special policies, tourism events and sudden disasters. The feature extraction of tourism UGC text from an abnormal time period can be used to analyze users' emotional trends.’

This approach will be the next step in our work and is currently already under study. We expect that the final results will also echo this paper well.

Your comments played an important role in improving the manuscript. Thank you for your constructive suggestions.

Please refer to lines 818–821 on page 36–37 of the revised ‘Manuscript’ and ‘Revised Manuscript with Track Changes’.

Reviewer #3: 

Comment 1:

It is better to add a new section “Literature Review” or “Existing Work” to summary previous research on related method of semantic knowledge discovery in GIScience, related spatiotemporal data mining method, tourism attraction analysis, granular computing model, etc. And then reorganize the section of Introduction.

Revision:

Thank you very much for your constructive comment.

In this revision, we added a "Literature Review" section that includes information on semantic knowledge discovery in GIScience, spatiotemporal data mining methods, tourism attraction analysis, and granular computing model. We also modified the introduction appropriately.

Your comments played an important role in improving the manuscript. Thank you for your constructive suggestions.

Please refer to lines 128–191 on pages 6–8 of the revised ‘Manuscript’ and ‘Revised Manuscript with Track Changes’.

Comment 2:

 The paper claims 5 aspects of contributions in introduction section. In my opinion, some contributions are not significant enough. For example, the 4th item “TAMTGC is extensible” cannot be thought of as a contribution. And Item 1 and 2 can be combined to illustrate the contribution of TAMTGC. Item 5 should be modified to claim the TAMTGC model was successfully applied in Jiuzhaigou area to obtain some new insightful research conclusion of tourist attractions in this area.

Revision:

Thank you very much for your constructive comment.

We have modified this part as follows.

The main contributions of this paper to tourism GIScience are as follows. (1) We introduce the granular computing (GrC) model into tourism geography through the TDPMTGC algorithm, which constructs a quantitative model of tourist destination popularity (TDP) at multi-spatiotemporal scales based on GrC using the inclusion degree. The proposed TDPMTGC can describe the TDP at a single spatial or temporal scale as well as the patterns and processes of TDP at multi-spatiotemporal scales. (2) A dataset construction approach for the text GrC model is proposed to provide a feasible scheme for reorganizing large-scale unstructured text and constructing public spatiotemporal UGC tourism datasets. (3) The TDPMTGC model was successfully applied in the Jiuzhaigou area, resulting in some new insightful conclusions regarding TDP in this area. TDPMTGC provides a new data mining approach for exploring tourist behaviors and analyzing the driving mechanisms of tourism patterns and processes both spatially and temporally.

Your comments played an important role in improving the manuscript. Thank you for your constructive suggestions.

Please refer to lines 113–123 on page 5 of the revised ‘Manuscript’ and ‘Revised Manuscript with Track Changes’.

Comment 3:

In Section 3, some formulas are very long and not very readable. Especially, in some sentences, some formulas have to be inserted, which makes readers confusing. For example, “the total number of A, B and C of 49 scenic spots”, and “the attraction of A of Zhenzhutanpubu is XXXXX”. A suggestion is, the authors can replace some formulas with simple symbols (use letters A, B, C, or use simple words), and use these simple symbols in sentences when complex formulas have to appear.

Revision:

Thank you very much for your constructive comment.

We agree that the formulas that were inserted in some sentences might confuse readers; therefore, we have substituted some simple symbols to replace these formulas in sentences. The upper right corner indicates the scale. When these letters appear in the text, the corresponding formula is used to calculate the popularity value. The modifications are as follows (the subsequent material are selected excerpts from Section 4.1.1 and 4.2.1):

(1) Scenic spot scale : a single spot, one route with multiple spots, multiple routes with multiple spots, expressed as , and , for simplicity, we use A4, B4, and C4 instead of , and , respectively, in the following passage. The calculation formulas are as follows:

A4: ,

B4: ,

C4: .

(2) Tourist route scale : we now use A3, B3, C3, D3 and E3 instead of , , , and , respectively. The calculation formulas are as follows:

A3: ,

B3: ,

C3: ,

D3: , and

E3: .

(3) Scenic area scale: we now use A2, B2, C2, D2 and E2 instead of , , , , and , respectively. The calculation formulas are as follows:

A2: ,

B2: ,

C2: ,

D2: , and

E2: .

Your comments played an important role in improving the manuscript. Thank you for your constructive suggestions.

Please refer to lines 415–441 on pages 18–19 and lines 488–565 on pages 21–25 of the revised ‘Manuscript’ and ‘Revised Manuscript with Track Changes’.

Comment 4:

In Section 4.2, the result of spatial scale is described using table including different place names as rows. It would be better to use maps to obtain better result visualization effects. Especially, most readers are not familiar with where Jiuzhaigou is, and where the locations of different tourist spots are. So a map of Jiuzhaigou describing locations of different travel spots could be helpful.

Revision:

Thank you very much for your constructive comment.

We apologize for ignoring the fact that most readers may not be familiar with the location of Jiuzhaigou or the locations of the different mentioned tourist spots. Therefore, we replaced Fig 3 with Fig 5 in Section 5.2 with the spatial distribution of Jiuzhaigou. In Fig 5, we divided the scenic spots into four levels according to their popularity value and marked the distribution of the popularity level of each scenic spot on the corresponding position of the route to which it belongs. Readers can easily find that scenic spots at different levels of spatiotemporal scale show different popularity distribution rules. This map not only includes the distribution rules for the tourist destination popularity of scenic spots within the route as shown in Fig 5, although it also includes the distribution location of each scenic spot, allowing readers to obtain the information more intuitively.

Your comments played an important role in improving the manuscript. Thank you for your constructive suggestions.

Please refer to 620-622 on page 28 of the revised ‘Manuscript’ and ‘Revised Manuscript with Track Changes’.

Comment 5:

From Table 2-5, it could be found that most calculation results are VERY small between 0.0000 and 0.0100. Can the authors consider some data normalization method, to normalize the intermediate data and final results to a value between 0.0 and 1.0, or a tourist attraction score between 0.0 and 100.0?

Revision:

Thank you very much for your constructive comment.

The difference in landscape popularity (i.e., the proportion of toponym text in the scenic area scale (93.33%) is much higher than that of nontoponym text (6.67%)) and the excessive number of landscape features (i.e., the number of scenic spots is 49, which leads to a smaller popularity value) resulted in popularity values between 0.0000 and 0.0100. In this revision, we normalized the numbers in Table 2-5 to keep them with the range 0-100%.

Your comments played an important role in improving the manuscript. Thank you for your constructive suggestions.

Please refer to lines 608–611 on pages 27–28 of the revised ‘Manuscript’ and ‘Revised Manuscript with Track Changes’.

Comment 6:

Some word and grammar errors can be found. There is a logic error in the FIRST sentence of this paper. It should be “tourism GIScience mainly studies a series of basic problems in XXXXX …” During my review of this paper, more than 10 grammar errors were found, including tense inconsistency and preposition errors. In addition, “multi-spatiotemporal” should be used instead of “multi-spatiotemporal”. When using “multi” with other nouns, there should always be a “-” between them.

Revision:

Thank you very much for your constructive comment.

We apologize for having made so many mistakes in writing the original paper. We have checked the entire text carefully and corrected the errors in the text (the modified part is marked with red font). If there are any more problems, please do not hesitate to let us know and we will correct them in time. We appreciate your help and support.

Your comments played an important role in improving the manuscript. Thank you for your constructive suggestions.

Please refer to the full revised ‘Manuscript’ and ‘Revised Manuscript with Track Changes’.

---

## [Decision Letter · Decision Letter 1]

9 Jan 2020

Measuring multi-spatiotemporal scale tourist destination popularity based on text granular computing

PONE-D-19-21928R1

Dear Dr. Renjie,

We are pleased to inform you that your manuscript has been judged scientifically suitable for publication and will be formally accepted for publication once it complies with all outstanding technical requirements.

With kind regards,

Professor Song Gao, Ph.D.

Academic Editor

PLOS ONE

Reviewers' comments:

Reviewer's Responses to Questions

**Comments to the Author**

1. If the authors have adequately addressed your comments raised in a previous round of review and you feel that this manuscript is now acceptable for publication, you may indicate that here to bypass the “Comments to the Author” section, enter your conflict of interest statement in the “Confidential to Editor” section, and submit your "Accept" recommendation.

Reviewer #1: All comments have been addressed

Reviewer #2: All comments have been addressed

Reviewer #3: All comments have been addressed

2. Is the manuscript technically sound, and do the data support the conclusions?

Reviewer #1: Yes

Reviewer #2: Yes

Reviewer #3: Yes

3. Has the statistical analysis been performed appropriately and rigorously? 

Reviewer #1: Yes

Reviewer #2: Yes

Reviewer #3: Yes

4. Have the authors made all data underlying the findings in their manuscript fully available?

Reviewer #1: Yes

Reviewer #2: Yes

Reviewer #3: (No Response)

5. Is the manuscript presented in an intelligible fashion and written in standard English?

Reviewer #1: Yes

Reviewer #2: Yes

Reviewer #3: Yes

6. Review Comments to the Author

Reviewer #1: The authors have sufficiently addressed my comments. The current spatiotemporal model is more scientifically sound than the original one.

Reviewer #2: (No Response)

Reviewer #3: I am glad to see the authors revised the paper according to my reviews carefully and seriously. Language and grammar problems have been edited. A map of research area has been added, and a single section of literature review was added with sufficient work. The formulas were more readable, and the authors also used some data normalization (percentage) to their analysis results. The biggest highlight of this paper is in data analysis section, where the authors provided detail information to analyze their results so the contribution of this paper could be highlighted. Spatial and temporal scale have been also combined in data analytics with sufficient details. For the general quality of this paper, I recommend that this paper could be accepted.

7. PLOS authors have the option to publish the peer review history of their article (what does this mean?). If published, this will include your full peer review and any attached files.

Reviewer #1: No

Reviewer #2: No

Reviewer #3: Yes: Kejin Cui

---

## [Editor Report · Acceptance letter]

18 Feb 2020

PONE-D-19-21928R1 

Measuring multi-spatiotemporal scale tourist destination popularity based on text granular computing 

Dear Dr. Renjie:

I am pleased to inform you that your manuscript has been deemed suitable for publication in PLOS ONE. Congratulations! Your manuscript is now with our production department. 

With kind regards,

on behalf of

Dr. Song Gao 

Academic Editor

PLOS ONE